# An intranasal nanoparticle STING agonist protects against respiratory viruses in animal models

Ankita Leekha[1], Arash Saeedi[1], Monish Kumar[1], K. M. Samiur Rahman Sefat[1], Melisa Martinez-Paniagua[1], Hui Meng[2], Mohsen Fathi[1], Rohan Kulkarni[1], Kate Reichel[1], Sujit Biswas[3], Daphne Tsitoura[4], Xinli Liu[3], Laurence J. N. Cooper[4], Courtney M. Sands[5], Vallabh E. Das[2], Manu Sebastian[4], Brett L. Hurst ⓞ[6] & Navin Varadarajan ⓞ[1] ✉

Respiratory viral infections cause morbidity and mortality worldwide. Despite the success of vaccines, vaccination efficacy is weakened by the rapid emergence of viral variants with immunoevasive properties. The development of an off-the-shelf, effective, and safe therapy against respiratory viral infections is thus desirable. Here, we develop NanoSTING, a nanoparticle formulation of the endogenous STING agonist, 2′−3′ cGAMP, to function as an immune activator and demonstrate its safety in mice and rats. A single intranasal dose of NanoSTING protects against pathogenic strains of SARS-CoV-2 (alpha and delta VOC) in hamsters. In transmission experiments, NanoSTING reduces the transmission of SARS-CoV-2 Omicron VOC to naïve hamsters. NanoSTING also protects against oseltamivir-sensitive and oseltamivir-resistant strains of influenza in mice. Mechanistically, NanoSTING upregulates locoregional interferon-dependent and interferon-independent pathways in mice, hamsters, as well as non-human primates. Our results thus implicate NanoSTING as a broad-spectrum immune activator for controlling respiratory virus infection.

Within the last 20 years, we experienced four global respiratory epidemics/pandemics: severe acute respiratory syndrome (SARS) in 2003, influenza H1N1 in 2009, Middle East respiratory syndrome coronavirus in 2012, and severe acute respiratory syndrome coronavirus 2 (SARS-CoV-2) in 2019. These pandemics added to the global burden of existing threats like seasonal Influenza and respiratory syncytial virus (RSV)[1,2]. The recent COVID-19 pandemic caused by SARS-CoV-2 has led to 6.17 million deaths (April 2022), while current outbreaks driven by new variants of concern (VOCs) continue to be reported worldwide. Three classes of interventions comprise the modern arsenal of responses against respiratory viruses: vaccines, antibodies, and antivirals[3–5]. All these three interventions require significant time for identification and characterization of the virus, development, and rapid testing to identify the emerging pathogen, followed by manufacturing and global distribution of therapeutics or vaccines. With regards to rapidly mutating viruses such as RNA viruses, all these three modalities are prone to failure due to the high mutation rate of the virus coupled with insufficient and ineffective protection, which facilitates the evolution of resistant variants[6–8].

Respiratory viruses enter the body and initiate replication in the respiratory tract. In response to the initial infection, the host elicits a multi-faceted innate immune response, typically characterized by the antiviral interferon (IFN) response, and the ensuing battle between the host immune system and the virus dictates the progression and

[1]William A. Brookshire Department of Chemical and Biomolecular Engineering, University of Houston, Houston, TX, USA. [2]College of Optometry, University of Houston, Houston, TX, USA. [3]Department of Pharmacological and Pharmaceutical Sciences, College of Pharmacy, University of Houston, Houston, TX, USA. [4]AuraVax Therapeutics, Houston, TX, USA. [5]Animal Care Operations, University of Houston, Houston, TX, USA. [6]Institute for Antiviral Research, Utah State University, Logan, UT, USA. ✉e-mail: nvaradar@central.uh.edu

outcome of infection[9,10]. Despite the IFN antagonistic mechanisms evolved by pathogens, these innate immunity responses dominate in most individuals and result in primarily asymptomatic infection or localized illness in the airways, still permitting an onward transmission of the virus[11,12]. If the host's innate immune response is suboptimal for any reason, including genetic defects or autoantibodies against IFNs, the viral infection progresses, leading to disseminated disease and even mortality[13,14]. Ensuring robust antiviral innate immune responses in the airways is central to controlling viral infection, replication, transmission, and disease outcomes. Although conceptually straightforward, harnessing this host antiviral response is challenging. Direct administration of IFN proteins in clinical trials for COVID-19 has yielded mixed results with undesirable side effects[15,16]. It is thus clear that the location, duration, and timing of host-directed immunotherapies are necessary to ensure the activation of the appropriate antiviral pathways that balance efficacy without causing tissue damage and toxicity.

The stimulator of the interferon genes (STING) pathway is an evolutionarily-conserved cellular sensor of cytosolic double-stranded DNA (dsDNA), enabling a broad innate immune response against viruses[17,18]. Mechanistically, activation of STING fosters an antiviral response that involves not just the type I and III interferons (IFN-I and IFN-III) but also additional pathways independent of interferon signaling[19,20]. In humans, pre-activated STING-mediated immunity in the upper airways controls early SARS-CoV-2 infection in children and can explain why children are much less susceptible to advanced disease[21,22]. Multiple reports have demonstrated that supra-physiologic activation of STING inhibits replication of viruses, including coronaviruses, and that viruses have evolved mechanisms to prevent the optimal activation of STING within the host[22,23].

Here, we demonstrate that intranasal delivery of a nanoparticle formulation of cyclic guanosine monophosphate–adenosine monophosphate (cGAMP), termed NanoSTING, enables the sustained release of cGAMP to both the nasal compartment and the lung for upto 48 h. We tested the ability of NanoSTING to protect against multiple VOCs of SARS-CoV-2 in hamsters and various variants of influenza A in mice. In these animal models, NanoSTING treatment prevented clinical disease, improved survival, reduced viral titers by several orders of magnitude, reduced transmission, and enabled durable protection from reinfection. In non-human primates, single- and repeat-dose administration of NanoSTING activated innate immunity in the nasal compartment. The stability, ease of administration, and the comprehensive nature of the immune response elicited make NanoSTING a promising, broad-spectrum antiviral, independent of the type of respiratory virus and variants.

## Results

### Preparation, characterization, and stability of NanoSTING
NanoSTING is a negatively charged liposomal formulation encapsulating endogenous STING agonist, cGAMP, optimized for the delivery of cGAMP to the respiratory tract (Fig. 1A)[24–26]. The composition of the lipids in our liposomal formulation has been shown to promote delivery to alveolar macrophages, facilitating the initiation of innate immune responses in the upper airways and the lung[27,28]. Dynamic light scattering (DLS) analysis revealed that the mean particle diameter of NanoSTING was 100 nm, with a polydispersity index of 23.6% (Supplementary Fig. 1A). The zeta potential of NanoSTING was −47 mV (Supplementary Fig. 1B). We confirmed the ability of NanoSTING to induce interferon responses by using THP-1 monocytic cells modified to conditionally secrete luciferase downstream of an Interferon regulatory factor (IRF) responsive promoter (Supplementary Fig. 1C). We stimulated THP-1 dual cells with NanoSTING at doses ranging from 2.5 to 10 µg and performed kinetic measurements for 24 h by measuring the luciferase activity in the supernatant. We observed a low level of luciferase activity at 6 h, and secretion was maximal at 24 h with 5 µg and 10 µg NanoSTING (Supplementary Fig. 1D). To evaluate the impact

of NanoSTING on cell viability, we measured the change in the percentage of dead cells after treatment using dynamic live cell imaging. Tracking the difference in the percentage of dead cells after 12 h of treatment with 2.5–10 µg of either cGAMP or NanoSTING revealed that stimulation by NanoSTING did not impact cell viability (Supplementary Fig. 1E). We next systematically measured the stability of the nanoparticles by assessing particle sizes and zeta potential of NanoSTING at two different temperatures, 25 °C, and 37 °C. While the hydrodynamic diameter of NanoSTING was essentially unchanged at 25 °C over a period of 30 days (Supplementary Fig. 1F), there was a slight increase in hydrodynamic diameter at 37 °C after 2 weeks (mean: 114 nm at 25 °C and 154 nm at 37 °C) [Supplementary Fig. 1G]. We did not observe a change in zeta potential at either temperature (−45 mV at 25 °C and 37 °C) [Supplementary Table 1]. These results demonstrate that NanoSTING remains stable even without refrigeration.

### NanoSTING delivers cGAMP across mucosa, leading to sustained Interferon-beta (IFN-β) secretion in the nasal compartment
Although cGAMP is a potent natural activator of STING, its clinical utility is hampered by a lack of cellular penetration and rapid degradation by plasma ectonucleotide pyrophosphatase phosphodiesterase 1 (ENPP1), leading to an in vivo half-life of only ~35 min[29]. We first characterized the ability of NanoSTING to mediate the delivery of cGAMP in the nasal compartment of mice. We delivered varying amounts of NanoSTING (10–40 µg) intranasally to groups of BALB/c mice, harvested the nasal turbinates and lungs, and assayed cGAMP using quantitative ELISA (Fig. 1A). We observed a dose-dependent increase in the concentration of cGAMP in the nasal turbinates; at the low dose (10 µg), we quantified cGAMP upto 12 h with a return to baseline at 24 h, whereas at the higher doses (20–40 µg), we detected cGAMP for 24 h with a return to baseline at 48 h (Fig. 1B). In the lungs, cGAMP was only detectable at the higher concentrations (20 and 40 µg) [Fig. 1C]. We also profiled the sera of these animals and observed that cGAMP was not detected at any time points in circulation, even at the highest dose (40 µg) [Supplementary Fig. 2]. These data confirmed that NanoSTING can transport cGAMP to the cells of the nasal passage in a concentration and time-dependent manner without systemic exposure.

The biological implications of NanoSTING's ability to deliver cGAMP and thus activate the STING pathway were evaluated using a panel of 10 genes to measure the immune response comprehensively. The panel comprised of the effector cytokines, C–X–C motif chemokine ligand 10 (Cxcl10) and interferon beta (Ifnb); Interferon stimulated genes (ISG) including Isg15, Interferon regulatory factor 7 (Irf7), myxovirus resistance proteins 1 and 2 (Mx1 and Mx2), and Interferon-induced protein with tetratricopeptide repeats 1 (Ifit1); and non-specific pro-inflammatory cytokines (Il6 and Tnf). BALB/c mice received varying doses of intranasal NanoSTING, and quantitative qRT-PCR was performed on the nasal turbinates (6–48 h) [Fig. 1A]. The effector cytokines Cxcl10 and Ifnb showed maximal induction (7000–20,000-fold) that remained elevated at 48 h (Fig.1D, E). The five ISGs demonstrated strong induction from 6 h (300–1000-fold) to 24 h, followed by a decline from 24 to 48 h (Fig. 1F–I and Supplementary Fig. 3). NanoSTING's inflammatory response was linked to the IFN pathway as the pro-inflammatory cytokine Il6 showed brief induction at 6 h (5000-fold), declined significantly by 24 h, and returned to baseline levels at 48 h (Fig. 1J). Furthermore, Tnf and Il10 showed only weak induction (15–60-fold) [Fig. 1K, L]. To rule out non-specific inflammation as the reason for the Ifnb1 responses in nasal turbinates, we intranasally administered groups of mice with liposomes without encapsulated cGAMP (Supplementary Fig. 4A, B). The Ifnb1 responses in animals administered with liposomes without encapsulated cGAMP were 100-fold lower than those in animals administered with NanoSTING, confirming that cGAMP is required for the robust induction of Ifnb1 (Supplementary Fig. 4C). Collectively, these results

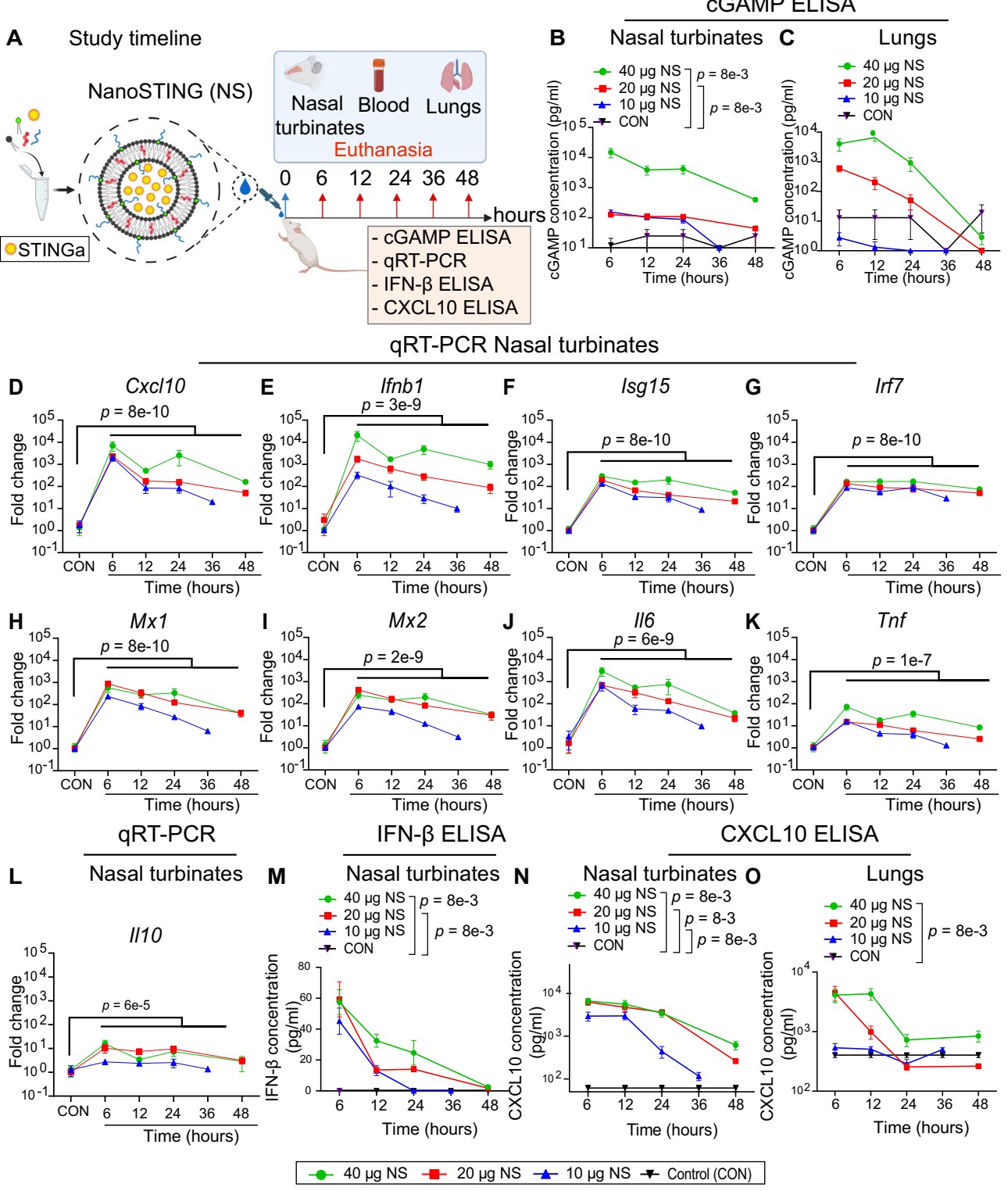

demonstrate that NanoSTING elicits a rapid and sustained inflammatory response triggering both effector cytokines and ISGs, but only minimal activation of non-specific pro-inflammatory cytokines.

Since the qRT-PCR data suggested strong induction of the effector cytokines, *Ifnb* and *Cxcl10*, we quantified the concentration of IFN-β and CXCL10 proteins in the nasal turbinates. Consistent with the transcriptional data, quantitative ELISA confirmed that both IFN-β and CXCL10 could be detected in the nasal turbinates and lungs for up to 24 h (Fig. 1M–O). We also tested the sera of the same animals. We did not observe either IFN-β or CXCL10 (Supplementary Fig. 2), confirming

that the stimulation of innate immunity by intranasal NanoSTING was localized in the airways without associated induction of systemic pro-inflammatory activity.

**Cellular targets of NanoSTING-mediated activation**

To track the cellular targets of NanoSTING, we synthesized liposomes by encapsulating sulphorhodamine (SRB), a red fluorescent dye with a charge and size similar to cGAMP. After synthesizing the liposomes, we conjugated them to DiD, a green fluorescent lipophilic carbocyanine dye. We dosed groups of mice intranasally with these dual-colored

**Fig. 1 | Pharmacokinetic and pharmacodynamic profiling of NanoSTING reveals prolonged delivery of cGAMP and induction of ISGs in the nasal compartment of mice. A** Overall schematic for the synthesis of NanoSTING and intranasal delivery of NanoSTING to mice. Groups of 3–12 *BALB/c* mice were treated with single doses of NanoSTING (10 μg, 20 μg, or 40 μg) and we euthanized subsets at 6 h, 12 h, 24 h, 36 h, and 48 h followed by collection of blood, nasal turbinates, and lungs. cGAMP ELISA, IFN-β ELISA, CXCL10 ELISA, and qRT-PCR (nasal turbinates) were the primary readouts. **B, C** ELISA quantification of cGAMP in the nasal turbinates and lungs of mice after treatment with NanoSTING. **D–L** Fold change in gene expression for NanoSTING-treated (40 μg in green, 20 μg in red, and 10 μg in blue) mice and control mice were quantified using RNA extracted from nasal turbinates by qRT-PCR (Primer sequences are provided in Supplementary Table 2). **M** Quantification of IFN-β concentration in mouse nasal tissue using quantitative ELISA. **N, O** Quantification of CXCL10 levels in mouse nasal tissue and lungs using quantitative ELISA. Individual data points represent independent biological replicates taken from separate animals; vertical bars show mean values with error bars representing SEM. Each dot represents an individual mouse. *P*-values were calculated by a two-tailed Mann–Whitney U-test for (**B–O**) ****$p < 0.0001$; ***$p < 0.001$; **$p < 0.01$; *$p < 0.05$; ns not significant. Data presented as combined results from (**B–O**) one independent animal experiment. Gender was not tested as a variable, and only female mice were included in the study. See also Supplementary Figs. 1–3 and Supplementary Table 2. Color codes: 40 μg NanoSTING (green), 20 μg NanoSTING (red), 10 μg NanoSTING (blue) and Control (black). **A** Created with BioRender.com released under a Creative Commons Attribution-NonCommercial-NoDerivs 4.0 International license (https://creativecommons.org/licenses/by-nc-nd/4.0/deed.en). Number of animals used: n = 3–12/group. Source data are provided as a Source Data file.

liposomes, harvested the lung and nasal turbinates at 12 h, and analyzed single-cell suspensions using flow cytometry (Fig. 2A). The frequency of the cells that were DiD⁺SRB⁺ was higher in the nasal tissue (6.4 ± 0.9%) compared to the lung (0.5 ± 0.2%) [Fig. 2B, C]. In both tissues, we specifically focused on four major subtypes of cells: epithelial cells (CD45⁻EPCAM⁺CD31⁻), endothelial cells (CD45⁻CD31⁺), and two myeloid cell subsets with the phenotype (i) CD45⁺EPCAM⁻CD11b⁺CD11c⁻ and (ii) CD45⁺EPCAM⁻CD11c⁺CD11b⁻ [Fig. 2D].

Comparing the nasal turbinates and lungs, the frequency of the epithelial cells that were DiD⁺SRB⁺ was higher for nasal turbinates (14.6 ± 3%) than for the lungs (0.1 ± 0%). We further investigated the three major subsets of epithelial cells in the nasal turbinates that were DiD⁺SRB⁺: secretory cells showed the highest frequency of DiD⁺SRB⁺ (41 ± 2%), followed by basal cells (33 ± 2%) and ciliated cells (26 ± 1%) [Fig. 2E, F]. In both tissues, only a low frequency of DiD⁺SRB⁺ cells were endothelial cells (Fig. 2D). Within the myeloid cell populations, we observed a higher frequency of CD45⁺EPCAM⁻CD11b⁺CD11c⁻ cells in nasal tissue (76.2 ± 7%) compared to lung (41 ± 8%). Conversely, the frequency of CD45⁺EPCAM⁻CD11c⁺CD11b⁻ was notably higher in the lung (43 ± 6%) compared to nasal tissue (4 ± 1%) [Fig. 2D, E]. In aggregate, the flow cytometry data revealed that NanoSTING is preferentially distributed to the nasal compartment, and activates the myeloid cell populations and diverse epithelial cell subsets within the nasal compartment.

## Repeat-dose administration of NanoSTING is safe and well-tolerated in mice and rats

We first studied biodistribution by altering the transport volume of intranasally delivered NanoSTING. It has been previously demonstrated that lower volumes lead to more efficient delivery to the nasal passage while larger volumes facilitate delivery to the lung[30]. Intranasal administration of Evan's blue dye in low and high volumes (40 μL and 120 μL) resulted in staining of the nasal turbinates, lungs, and stomach in hamsters (Supplementary Fig. 6A–C). However, at both volumes, there was a significant amount of the dye delivered to the nasal turbinates and lung (intended target organs) [Supplementary Fig. 6A], and the normalized ratio of distribution to these tissues was independent of the volume of administration (Supplementary Fig. 6B, C). These results suggested that biodistribution to the lung/nasal compartments after intranasal delivery of liquid formulations was not impacted by the volume of inoculum.

To investigate if the liposomal nanoparticle formulation can lead to toxicity, we administered a single dose of nanoparticles (without encapsulated cGAMP) to mice. We harvested the lungs and stomach (Supplementary Fig. 7A). Histopathology of all these organs was unremarkable, confirming that the lipid components, when formulated as nanoparticles, are not toxic (Supplementary Fig. 7B). To investigate the toxicology of NanoSTING, we used allometric scaling based on both the body mass and nasal surface (intranasal delivery) to identify the appropriate doses for intranasal delivery to rats. We administered a low dose of 50 μg (low dose, equivalent to 10 μg in mice) and 250 μg (high dose, equivalent to 40 μg in mice) intranasally to groups of rats, performed routine clinical observations, and monitored the weight daily. There was no significant difference in food consumption, body weight changes, loss of fur, or any other clinical observations between the treatment and control groups for either sex. Similarly, hematology, coagulation, clinical chemistry, and urinalysis were normal in the NanoSTING-treated rats (Supplementary Tables 5–8).

Repeat-dose toxicity studies are vital to drug development and help identify potential toxicity in the appropriate target organs. We administered groups of rats four doses of NanoSTING (125 μg, mid-dose). We harvested the small intestines, stomach, lungs, and nasal cavity (Fig. 3A). Histopathology of all these organs was unremarkable, confirming that repeat-dose administration of NanoSTING is safe (Fig. 3B). Similarly, hematology, coagulation, clinical chemistry, and urinalysis were all normal upon repeat-dose administration of NanoSTING (Supplementary Tables 9–12). In aggregate, these results demonstrated that repeat-dose administration of NanoSTING did not affect any toxicological parameters in animals.

## RNA-sequencing confirms a robust IFN-I signature in the lungs of hamsters following intranasal NanoSTING administration

Next, we wanted to investigate the impact of intranasal NanoSTING administration on the lungs of Syrian golden hamsters (*Mesocricetus auratus*). The hamster is a well-characterized model for the SARS-CoV-2 challenge and mimics severe disease in humans; animals demonstrate easily quantifiable clinical disease characterized by rapid weight loss, very high viral titers in the lungs, and extensive lung pathology[31]. Additionally, unlike the K18-hACE2 transgenic model, hamsters recover from the disease (like most humans) and hence offer the opportunity to study the impact of treatments in the disease process and virus transmission[31,32].

To assess the impact of intranasal NanoSTING on the lung, we administered one group (n = 4/group) of hamsters with daily doses of NanoSTING (60 μg) for four consecutive days. We used naive hamsters as controls (n = 4/group). Both groups of hamsters showed no differences in clinical signs, such as temperature or body weight (Supplementary Fig. 8A, B). On day 5, we isolated the lungs from hamsters for unbiased whole-transcriptome profiling using RNA-sequencing (RNA-seq). At a false-discovery rate (FDR *q-value* < 0.25), we identified a total of 2922 differentially expressed genes (DEGs) between the two groups (Fig. 4A). A type I IFN response was induced in NanoSTING-treated lungs, comprising canonical ISGs, including *Mx1, Isg15, Uba7, Ifit2, Ifit3, Ifit35, Irf7, Adar*, and *Oas2* (Fig. 4B)[33]. The effector cytokines, *Cxcl9-11* and *Ifnb*, were also induced in treated hamsters (Fig. 4C) and showed robust induction of direct antiviral proteins, such as *Ddx60* and *Gadd45g* (Fig. 4D)[34,35]. We performed gene set enrichment analysis

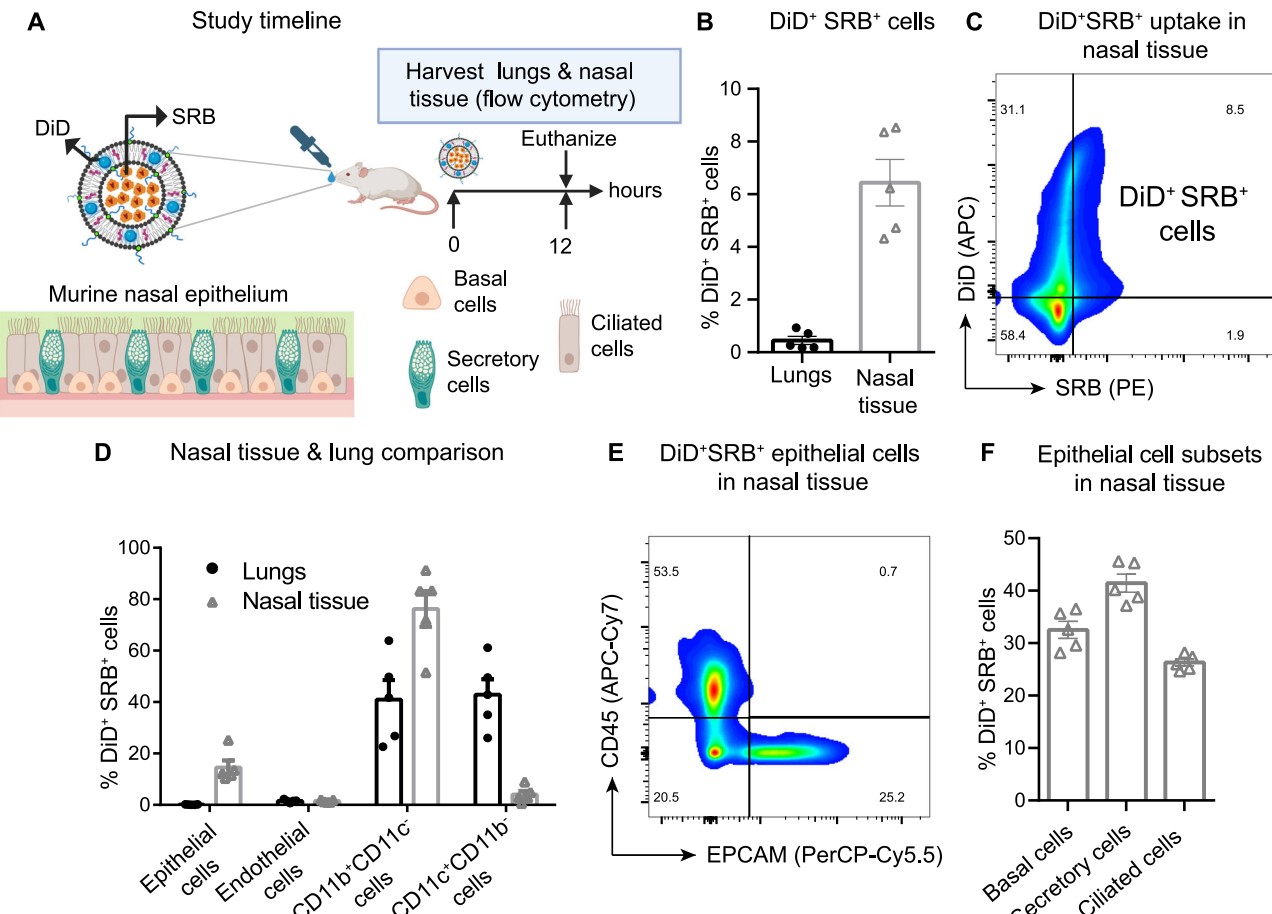

**Fig. 2 | Uptake of NanoSTING by myeloid populations and epithelial cells in nasal tissue and lungs. A** Overall schematic for tracking the cellular targets of NanoSTING. The liposomes were formulated to encapsulate SRB (red dye) and the liposomes were conjugated to DiD (green dye). The dual-labeled liposomes were administered intranasally to mice, and the single-cell suspensions were analyzed using flow cytometry. The cell types of the murine nasal epithelium are shown schematically. **B** Quantification of DiD⁺ SRB⁺ cells in lungs and nasal tissue by flow cytometry. **C** Flow cytometric plots (pseudocolor-smooth) showing uptake of DiD & SRB in nasal tissue. **D** Quantification of DiD⁺ SRB⁺ epithelial cells (CD45⁻EPCAM⁺), endothelial (CD45⁻CD31⁺), and two myeloid cell subsets-CD45⁺EPCAM⁻CD11b⁺CD11c⁻ and CD45⁺EPCAM⁻CD11c⁺CD11b⁻ in lungs and nasal tissues by flow cytometry. **E** Flow cytometric plots (pseudocolor-smooth) showing uptake of DiD & SRB by epithelial cells in nasal tissue. **F** The percentages of epithelial cells (basal cells, secretory cells & ciliated cells) in nasal tissue. Individual data points represent independent biological replicates taken from separate animals; vertical bars show mean values with error bars representing SEM. Each dot represents an individual mouse. Data presented as combined results from (**B**–**F**) one independent animal experiment. Gender was not tested as a variable, and only female mice were included in the study. See also Supplementary Fig. 5 (gating strategy), Supplementary Table 4 (list of antibodies or conjugates used). Color codes: Lungs (Black), Nasal tissue (gray). **A** Created with BioRender.com released under a Creative Commons Attribution-NonCommercial-NoDerivs 4.0 International license (https://creativecommons.org/licenses/by-nc-nd/4.0/deed.en). Number of animals used: n = 5/group. Source data are provided as a Source Data file.

(GSEA) to compare the differentially induced pathways upon treatment with NanoSTING. We interrogated the changes in these populations against the Molecular Signatures Database (Hallmark, C2, and C7 curated gene sets). We observed a distinct cluster of pathways related to both type I and type III interferons in the lungs of NanoSTING-treated mice. We confirmed the specificity of the response by qRT-PCR analyses by quantifying *Mx1-2*, *Isg15*, *Irf7*, *Cxcl11*, *Ifnb*, *Il6*, and *Il10* (Supplementary Fig. 9). Since the gene signature of interferon-independent activities of STING is known[19], we performed GSEA and confirmed that NanoSTING activates interferon-independent pathways (Fig. 4E, F). In aggregate, these results demonstrate that cGAMP-mediated activation of STING by NanoSTING efficiently engages both interferon-dependent and interferon-independent antiviral pathways in the lung.

### Quantitative modeling predicts that early treatment with NanoSTING will dampen viral replication

The in vivo mechanistic experiments demonstrated that NanoSTING induces a broad antiviral response by engaging the innate immune system. To investigate potential efficacy, we used a mathematical model in combination with human viral titer data to identify the treatment window and quantify the relative amount of type I IFN (or related pathways) elicited by NanoSTING required for therapeutic benefit[36,37]. To simplify the framework of the model, we assumed that in vivo cGAMP only works to stimulate interferon responses. With this assumption, we modeled the range of relative interferon ratios (RIR, 0–1) we need to elicit via NanoSTING in comparison to the population level peak interferon responses observed upon SARS-CoV-2 infection (Fig. 5A, B) (equation 1 and 2) [Refer to Sup Note 1] and investigated the influence on viral elimination. Based on the model, an RIR of just 0.27 (27% of natural infection) would be sufficient to achieve a 50% reduction in viral titer (based on the area under the curve, AUC), and RIR values of at least 0.67 would achieve a 100% reduction in viral titers (Fig. 5C). We next modeled the window of initiation of treatment, which revealed that intervention would be most effective when initiated within 2 days after infection (Fig. 5D). By contrast, if the treatment is initiated after the peak of viral replication, even with an RIR of 1, improvement in outcomes cannot be readily realized (Fig. 5D and

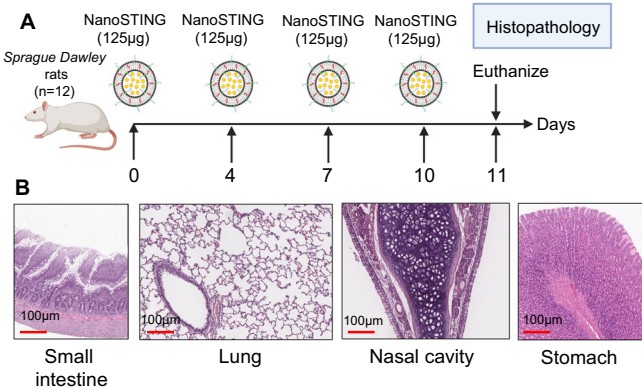

**Fig. 3 | Rat toxicology studies based on repeat-dose administration of NanoSTING. A** Groups of *Sprague Dawley* rats (n = 12) were intranasally administered four doses of 125 μg on the indicated days and euthanized on day 11 for histopathology of the small intestines, lungs, nasal cavity, and stomach. **B** Representative H & E images of the target organs of the treated rats; all images were acquired at 10×; scale bar, 100 μm. Gender was tested as a variable with an equal number of male and female rats included in the study. See Supplementary Figs. 6, 7. **A** Created with BioRender.com released under a Creative Commons Attribution-NonCommercial-NoDerivs 4.0 International license (https://creativecommons.org/licenses/by-nc-nd/4.0/deed.en). Number of animals used: n = 12/group. Source data are provided as a Source Data file.

Supplementary Fig. 10C). Collectively, these results from quantitative modeling predicted that: (i) a single dose of NanoSTING is adequate to elicit only a moderate amount of IFN which is likely achievable since our data supports large induction of IFN-β (Figs. 1E, M, and 4F) and given that natural infections with viruses like SARS-CoV-2 and Influenza A are known to suppress interferon production[38–40], and (ii) the optimal treatment window was either as prophylaxis treatment or initiated early after infection.

**NanoSTING protects against the SARS-CoV-2 Delta VOC**
Based on the prediction of the modeling studies, we evaluated whether a single dose of NanoSTING protects hamsters from SARS-CoV-2 infection. The SARS-COV2 Delta VOC (B.1.617.2) was chosen because it causes upper- and lower-respiratory tract diseases and has increased disease severity compared to prior VOCs (Wuhan and Beta strains).[41] We treated groups of 12 Syrian golden hamsters with a single intranasal dose of 120 μg NanoSTING, and 24 h later, we infected the hamsters with ~3 × 10⁴ 50% Cell culture infectious dose ($CCID_{50}$) of the Delta VOC via intranasal route (Fig. 6A). In the placebo-treated (PBS) group, hamsters exhibited weight loss, with a mean peak weight loss of 8 ± 2%. By contrast, hamsters treated with NanoSTING were largely protected from weight loss (mean peak weight loss of 2.7 ± 0.7%) (Fig. 6B, C). This small amount of weight loss in hamsters was similar to the results obtained by adenoviral vectored vaccines challenged with either the Wuhan or Beta strains[42]. We quantified the infectious viral titers by sacrificing six hamsters on day 2. Even with the highly infectious Delta VOC, NanoSTING reduced infectious viral titers in the lung post-two days of infection by 300-fold compared to placebo-treated animals (Fig. 6D). This reduction in lung viral titers closely correlates with weight loss prevention in these animals and models protection similar to clinical human disease. We also quantified the viral titers in the nasal compartment. We observed that treatment with NanoSTING reduced infectious viral titers in the nasal compartment post-two days of infection by 1000-fold compared to placebo-treated animals (Fig. 6E). The reduction in viral replication in the nasal compartment models the propensity of human transmission and confirms that treatment with NanoSTING decreases the likelihood of transmission. To map the duration of efficacy of prophylactic NanoSTING treatment, we

administered a single intranasal dose of NanoSTING (120 μg) and challenged the hamsters 72 h later with ~3 × 10⁴ $CCID_{50}$ of the Delta VOC (Supplementary Fig. 11A). Even when administered 72 h before exposure, NanoSTING showed moderate protection from weight loss and a significant reduction in infectious viral titers (Supplementary Fig. 11B–D). Our model also predicts that NanoSTING can be used to control infection after viral exposure. To test efficacy post-exposure, we delivered intranasal NanoSTING 6 h after exposure to the Delta VOC (Supplementary Fig. 12A). We observed a 340-fold and 13-fold reduction in infectious virus in the nasal passage and lung, respectively (Supplementary Fig. 12B, C). These results showed that a single-dose treatment with NanoSTING can effectively minimize clinical symptoms, protect the lungs, and reduce infectious viruses in the nasal passage.

**Treatment with NanoSTING induces protection against SARS-CoV-2 reinfection**
One of the advantages of enhancing innate immunity to clear a viral infection is that this process mimics the natural host defense and minimizes the danger of clinical symptoms. We hypothesized that NanoSTING treatment via innate immune system activation also facilitates immunological memory against reinfection without additional treatment. To test this hypothesis, we intranasally treated Syrian golden hamsters (n = 12/group) with NanoSTING (120 μg) and 24 h later challenged with ~3 × 10⁴ of the SARS-CoV-2 Delta VOC (B.1.617.2) (Fig. 6A). On day 28, we rechallenged the hamsters with the Delta VOC. The untreated animals suffered significant weight loss during the primary challenge but were largely protected during the secondary challenge (Fig. 6F). By contrast, NanoSTING-treated hamsters showed minimal weight loss during the primary challenge, which did not compromise immunological memory. Indeed, NanoSTING-treated hamsters were completely protected from weight loss during the secondary challenge, and their body weight was identical to animals that were not previously challenged (Fig. 6F). These results suggest that a single intranasal treatment with NanoSTING activates the antiviral program of innate immunity, preventing clinical disease during primary infection while offering durable protection from reinfection.

**NanoSTING treatment protects against the SARS-CoV-2 Alpha VOC**
We next evaluated the impact of treatment with varying doses of NanoSTING and varied the dose of treatment based on the duration of response that we have documented (Fig. 1). The SARS-COV2 Alpha VOC (B.1.1.7) is known to be resistant to IFN-1 signaling in vitro and thus provides a challenging model to test the efficacy of NanoSTING[43,44]. We pre-treated Syrian golden hamsters (n = 6/group) with two intranasal doses of NanoSTING (30 μg and 120 μg) and 24 h later challenged the hamsters with ~3 × 10⁴ $CCID_{50}$ of the Alpha VOC (Fig. 6G). Treatment with either dose of NanoSTING protected the hamsters from severe weight loss (Fig. 6H). We used an integrated scoring rubric (range from 1 to 12) that accounts for the histopathology of the lung tissue on day six after the viral challenge. We observed that NanoSTING-treated hamsters had significant reductions in aggregate pathology scores with minimal evidence of invasion by inflammatory cells or alveolar damage (Fig. 6I, J). In addition, we quantified the viral titers in the lungs and nasal compartments. We observed a significant reduction of viral titers in both compartments as early as day two post-challenge (Fig. 6K, L). Thus, treatment with intranasal NanoSTING reduces in vivo replication of SARS-CoV-2 by orders of magnitude and confers protection against IFN-I evasive strains of SARS-CoV-2.

**NanoSTING treatment prevents infection in hamsters exposed to the SARS-CoV-2 Omicron VOCs**
The SARS-CoV-2 Omicron VOC (BA.5) is among the most infectious strains of SARS-CoV-2. Using the Omicron VOC sets a high bar for

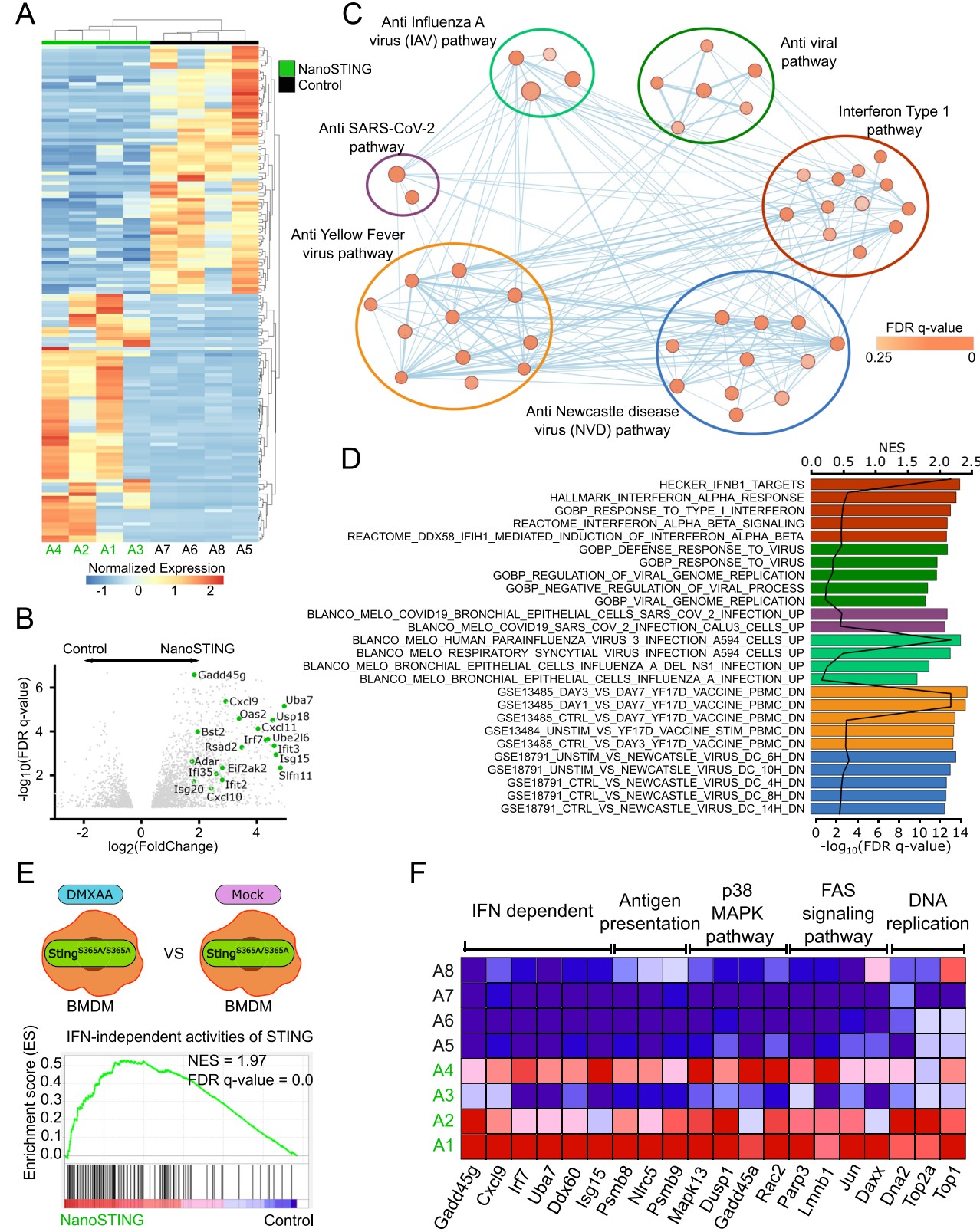

NanoSTING to prevent viral transmission. We set up a transmission experiment designed to answer two fundamental questions: (1) does the prophylactic treatment of infected (index) hamsters prevent transmission to contact hamsters, and (2) does the post-exposure treatment of contact hamsters mitigate viral replication? Accordingly, we set up an experiment with three groups (n = 15) of Syrian golden hamsters. In each group, five index hamsters were intranasally infected

with the ~3 × 10^4 CCID_50 of the SARS-CoV-2 Omicron VOC (BA.5). We quantified the viral titers in the infected and contact hamsters that were: (a) cohoused with placebo-treated infected index hamsters (group 1), (b) cohoused with NanoSTING (120 μg) treated index hamsters (group 2), or (c) treated with NanoSTING after cohousing with infected but untreated hamsters (group 3) [Fig. 7A]. The animals were co-housed continuously for 4 days such that transmission could

**Fig. 4 | RNA-sequencing identifies the activation of IFN-dependent and IFN-independent pathways in the lungs of hamsters treated with NanoSTING.**
**A** Heatmap of the top 50 differentially expressed genes (DEGs) between NanoSTING-treated lungs (marked as green) and control lungs (marked as black). **B** The volcano plots of DEGs comparing NanoSTING-treated and control animals. **C** Gene set enrichment analyses (GSEA) of C2 and C7 curated pathways visualized using Cytoscape. Nodes (red and blue circles) represent pathways, and the edges (blue lines) represent overlapping genes among pathways. The size of nodes represents the number of genes enriched within the pathway, and the thickness of edges represents the number of overlapping genes. The color of nodes was adjusted to an FDR q-value ranging from 0 to 0.25. Clusters of pathways are labeled as groups with a similar theme. **D** The normalized enrichment score (NES) and false-discovery rate (FDR) *q values* of top antiviral pathways curated by GSEA analysis.

**E** GSEA of IFN-independent activities of STING pathway activated in the lung of NanoSTING-treated animals. The schematic represents the comparison that was made between samples collected from the GSE149744 dataset to generate the pathway gene set. **F** The expression of genes in lungs associated with IFN-dependent and IFN-independent antiviral pathways between NanoSTING and control groups. Data represents independent biological replicates taken from separate animals. Data presented as combined results from one independent animal experiment. Gender was tested as a variable with an equal number of male and female hamsters included in the study. See also Supplementary Figs. 8 and 9 and Supplementary Table 3. Color codes: Control (Black), NanoSTING (green). Fig. 1A-Created with BioRender.com released under a Creative Commons Attribution-NonCommercial-NoDerivs 4.0 International license (https://creativecommons.org/licenses/by-nc-nd/4.0/deed.en). Number of animals used: n = 4/group.

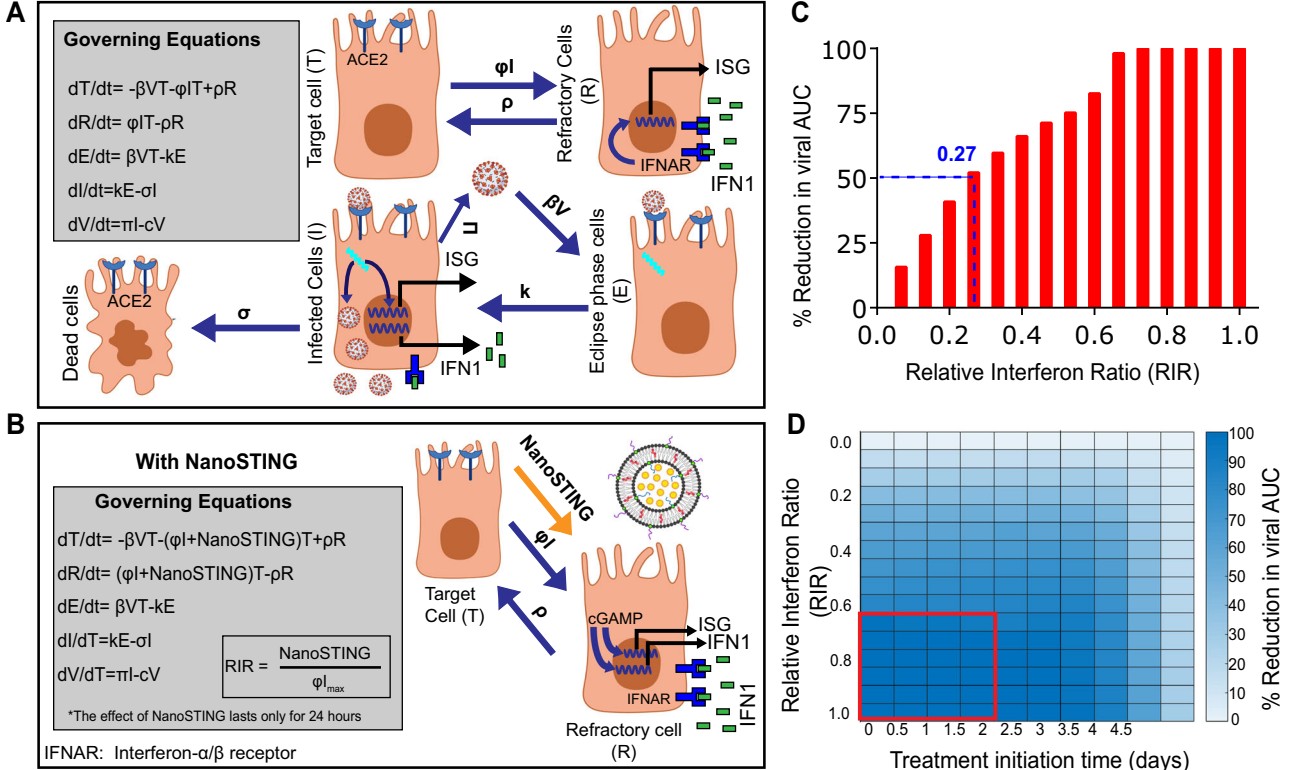

**Fig. 5 | Quantitative modeling of the dynamics of replication of SARS-CoV-2.**
**A**, **B** Schematic representing rate constants and equations governing viral dynamics during **A** natural infection and **B** in the presence of NanoSTING treatment. **C** Reduction in the viral area under the curve (AUC) at different NanoSTING efficacies (RIR) compared to natural infection. The treatment is initiated on day 0, and we assume that the effects of NanoSTING treatment only last for 24 h. **D** Heatmap of viral AUC with varying NanoSTING efficacy and treatment initiation time. The red box represents the combination with close to 100% reduction in viral AUC. See Supplementary Fig. 10, Supplementary Tables 13–15, and Supplementary Note 1. Fig. 5A, B Created with BioRender.com released under a Creative Commons Attribution-NonCommercial-NoDerivs 4.0 International license (https://creativecommons.org/licenses/by-nc-nd/4.0/deed.en).

happen through aerosols and also direct contact and fomite (including diet and bedding).

As with the other strains of SARS-CoV-2 that we tested, NanoSTING pre-treatment of the infected hamsters led to a 1000-fold and 160-fold decrease in the viral load in the lungs and nasal compartment on day five compared to untreated animals (Fig. 7B, C). The reduction in viral loads was accompanied by efficient prevention of transmission to cohoused but untreated hamsters (4/5 of animals were virus-free in the NanoSTING group compared to 0/5 virus-free in the untreated group) [Fig. 7B, C]. Significantly, post-exposure treatment of the contact hamsters was also effective at reducing viral titers, although the magnitude of reduction was smaller compared to the animals that were directly challenged with the virus (Fig. 7C).

We repeated these transmission studies with the SARS-CoV-2 Omicron VOC (B.1.1.529) (Fig. 8A). We observed that NanoSTING pre-treatment of the infected hamsters almost completely blocked transmission (7/8 animals treated were virus-free vs 1/8 untreated animals were virus-free) [Fig. 8B]. Significantly, post-exposure treatment of the contact hamsters was also effective at preventing infection (6/8 animals treated were virus-free), and all animals demonstrated a significant reduction in viral titers (Fig. 8C). Consistent with the known milder disease of the Omicron VOC, none of the infected animals showed weight loss (Supplementary Fig. 13)[45]. Collectively, these results directly demonstrate that NanoSTING treatment is highly effective at blocking transmission even with the highly infectious Omicron VOC.

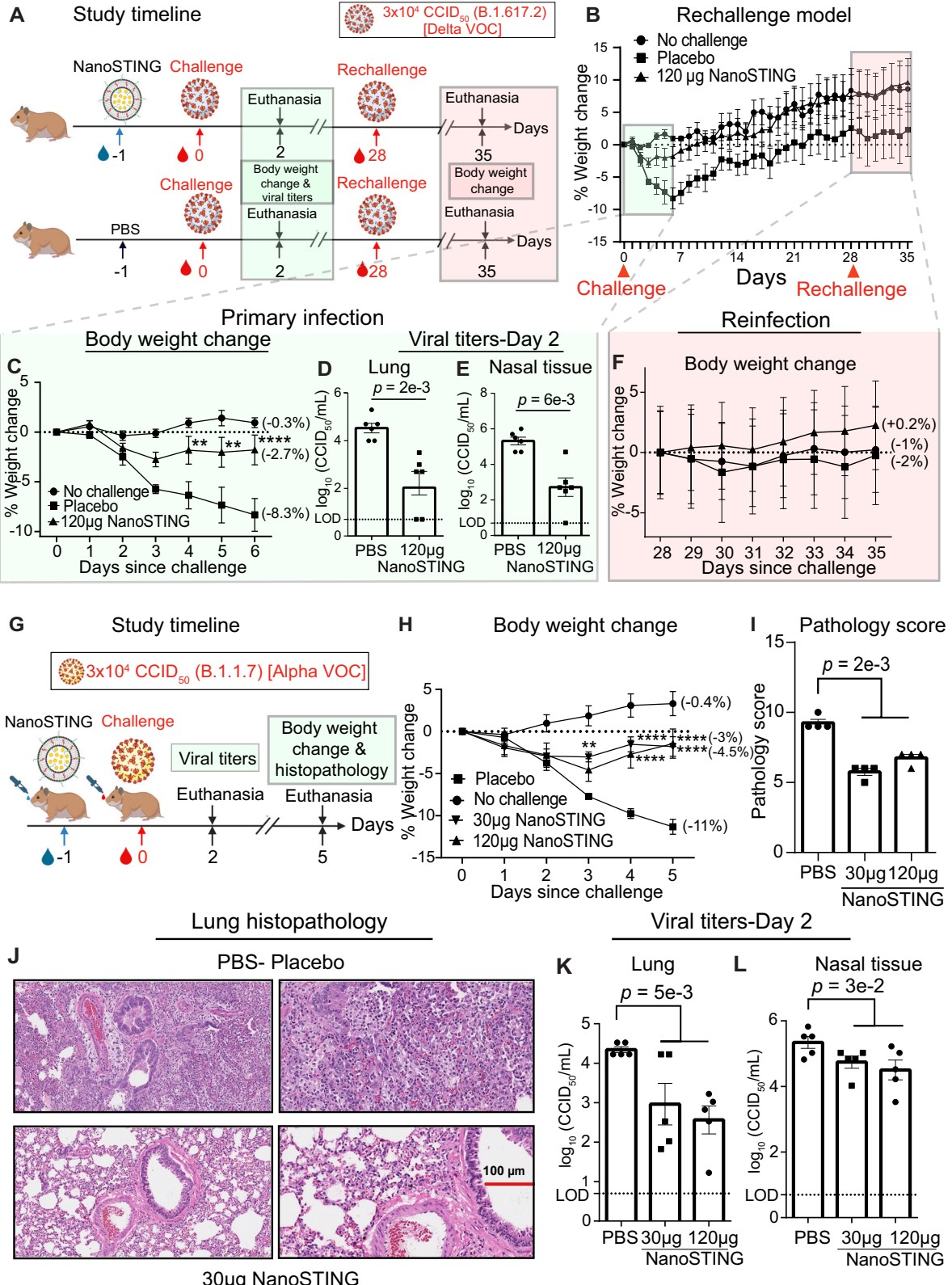

**A** Study timeline

3x10⁴ CCID₅₀ (B.1.617.2) [Delta VOC]

**B** Rechallenge model

**C** Primary infection — Body weight change

**D** Lung — Viral titers-Day 2
p = 2e-3

**E** Nasal tissue
p = 6e-3

**F** Reinfection — Body weight change

**G** Study timeline
3x10⁴ CCID₅₀ (B.1.1.7) [Alpha VOC]

**H** Body weight change

**I** Pathology score
p = 2e-3

**J** Lung histopathology — PBS- Placebo
100 µm
30µg NanoSTING

**K** Lung — Viral titers-Day 2
p = 5e-3

**L** Nasal tissue
p = 3e-2

## Treatment with NanoSTING induces protection from influenza superior to oseltamivir

Influenza viruses have evolved multiple mechanisms to dampen the host's innate immunity, including the attenuation of interferon responses by the NS1 protein[46,47]. One of the primary treatment options against influenza involves post-exposure treatment using oseltamivir, which inhibits the influenza neuraminidase protein. We

thus compared the efficacy of NanoSTING in comparison to oseltamivir in mouse models of influenza.

We challenged groups of ten mice with $2 \times 10^4$ CCID₅₀ of Influenza A/California/04/2009 (H1N1dpm). We treated them with a clinically relevant dose of oseltamivir (30 mg/kg/day) twice daily for five days (Supplementary Fig. 14A)[48]. The untreated animals started losing significant weight by day three and showed a mean peak weight loss of

**Fig. 6 | Protective efficacy of NanoSTING against the pathogenic SARS-CoV-2 Delta (B.1.617.2) VOC and IFN evasive SARS-CoV-2 Alpha VOC (B.1.1.7). A** We treated groups of 12 Syrian Golden hamsters, each with a single dose of 120 µg NanoSTING, and later challenged with ~3 × 10⁴ CCID₅₀ of SARS-CoV-2 Delta VOC on day 0 by the intranasal route. We euthanized half of the hamsters (n = 6) hamsters on day 2 and determined viral titers of lung and nasal tissues. We rechallenged the remaining 6 hamsters on day 28 and tracked the body weight change until day 35. **B** Percent body weight change compared to the baseline at the indicated time intervals. **C** Percent body weight change monitored during the primary infection (day 0–day 6). **D, E** Viral titers measured by endpoint titration assay in nasal tissues and lungs post-day 2 of infection. The dotted line indicates the limit of detection of the assay (LOD). **F** Percent body weight change monitored after rechallenge (day 28–day 35). **G** We tested groups of 9 hamsters, each with two different doses of NanoSTING (30 µg and 120 µg) and 24 h later challenged with the ~3 × 10⁴ CCID₅₀ of SARS-CoV-2 Alpha VOC (B.1.1.7). On day 2, five animals from each group were euthanized for assessing the viral titers and remaining animals used for the histo-pathology at day 5. No animals were excluded in this study. **H** Change in body weight of hamsters. **I, J** Pathology scores and a representative hematoxylin and eosin (H & E) image of the lung showing histopathological changes in lungs of hamsters treated with NanoSTING (30 µg) and PBS; all images were acquired at 10× and 20×; scale bar, 100 µm. **K, L** Viral titers were quantified in the lung and nasal tissue by endpoint titration assay on day 2 after the challenge. The dotted line

indicates the limit of detection of the assay (LOD). Individual data points represent independent biological replicates taken from discrete samples; vertical bars show mean values with error bars representing SEM. Each dot represents an individual hamster. For (**D, E, I, K, L**), analysis was performed using a two-tailed Mann–Whitney U-test. For (**C, H**), data was compared via a mixed-effects model for repeated measures analysis. Lines depict group mean body weight change from day 0; error bars represent SEM. Asterisks indicate significance compared to the placebo-treated animals at each time point. Mann–Whitney U-test ****$p < 0.0001$; ***$p < 0.001$; **$p < 0.01$; *$p < 0.05$; ns not significant. For (**B**), the p-values are as follows: Day 4: $p = 6e{-}3$, Day 5: $p = 1e{-}3$, and Day 6: $p = 5e{-}5$. For (**H**), the exact p-values comparing the 30 µg NanoSTING group to the Placebo group are Day 3: $p = 5e{-}3$, Day 4: $p = 6e{-}7$, and Day 5: $p = 10e{-}9$. Additionally, for the 120 µg NanoSTING and Placebo-treated group, the p-values are Day 4: $p = 2e{-}5$ and Day 5: $p = 3.5e{-}9$. Data presented as combined results from two independent experiments [**A–F** Challenge study with SARS-CoV-2 Delta VOC, **G–L** challenge study with SARS-CoV-2 Alpha VOC)], each involving one independent animal experiment. Gender was tested as a variable, and an equal number of male and female hamsters were included in the study. See also Supplementary Figs. 11 and 12. Figure 6A, G−Created with BioRender.com released under a Creative Commons Attribution-NonCommercial-NoDerivs 4.0 International license (https://creativecommons.org/licenses/by-nc-nd/4.0/deed.en). Number of animals used in the study: n = 12/group (for **A–F**), n = 9/group (for **G–L**). Source data are provided as a Source Data file.

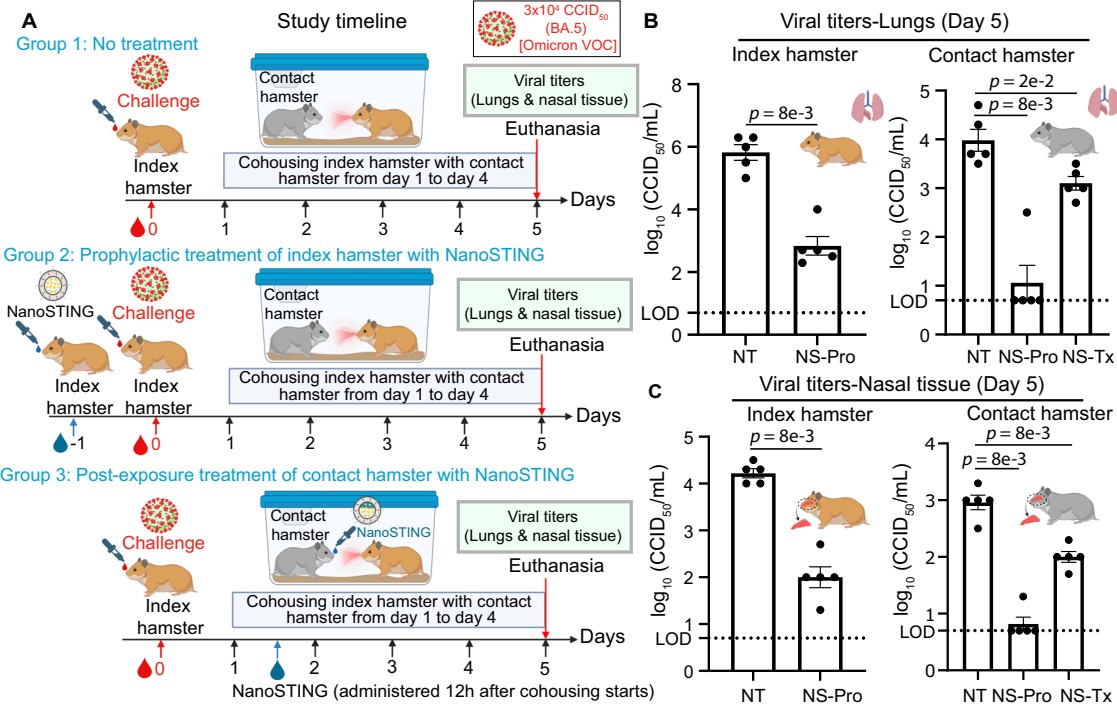

**Fig. 7 | Intranasal administration of NanoSTING limits transmission and viral replication in the lungs and nasal passage of contact hamsters exposed to the SARS-CoV-2 Omicron (BA.5) VOC. A** Experimental setup: For group 1, we challenged groups of 5 hamsters each on day 0 with ~3 × 10⁴ of SARS-CoV-2 Omicron VOC (BA.5) and after 24 h cohoused index hamsters in pairs with contact hamsters (n = 5) for 4 days in clean cages. In group 2, we pre-treated the hamsters with 120 µg of NanoSTING 24 h prior to infection. In group 3, we treated the contact hamsters with NanoSTING 12 h after the cohousing period began. We euthanized the contact and index hamsters on day 4 of cohousing. Viral titers in the nasal tissue of the index and contact hamsters were used as primary endpoints. **B** Viral titers were quantified in the lung of the index (infected) and contact hamsters by endpoint titration assay post-day 5 of infection. **C** Viral titers were quantified in the nasal tissue of index and contact hamsters by endpoint titration assay post-day 5 of infection. The dotted line indicates the limit of detection of the assay (LOD). For

(**B, C**) analysis was performed using a two-tailed Mann–Whitney U-test. Individual data points represent independent biological replicates taken from separate animals; vertical bars show mean values with error bars representing SEM. Each dot represents an individual hamster. Asterisks indicate significance compared to the placebo-treated animals. ****$p < 0.0001$; ***$p < 0.001$; **$p < 0.01$; *$p < 0.05$; ns not significant. Data presented as combined results from one (**B, C**) independent animal experiment. NT Non-treated, NS-Pro Prophylactic treatment with NanoSTING, NS-Tx Post-exposure treatment with NanoSTING. Gender was tested as a variable with an equal number of male and female hamsters included in the study. **A** and parts of **B, C**−Created with BioRender.com released under a Creative Commons Attribution-NonCommercial-NoDerivs 4.0 International license (https://creativecommons.org/licenses/by-nc-nd/4.0/deed.en). Number of animals used: n = 5/group. Source data are provided as a Source Data file.

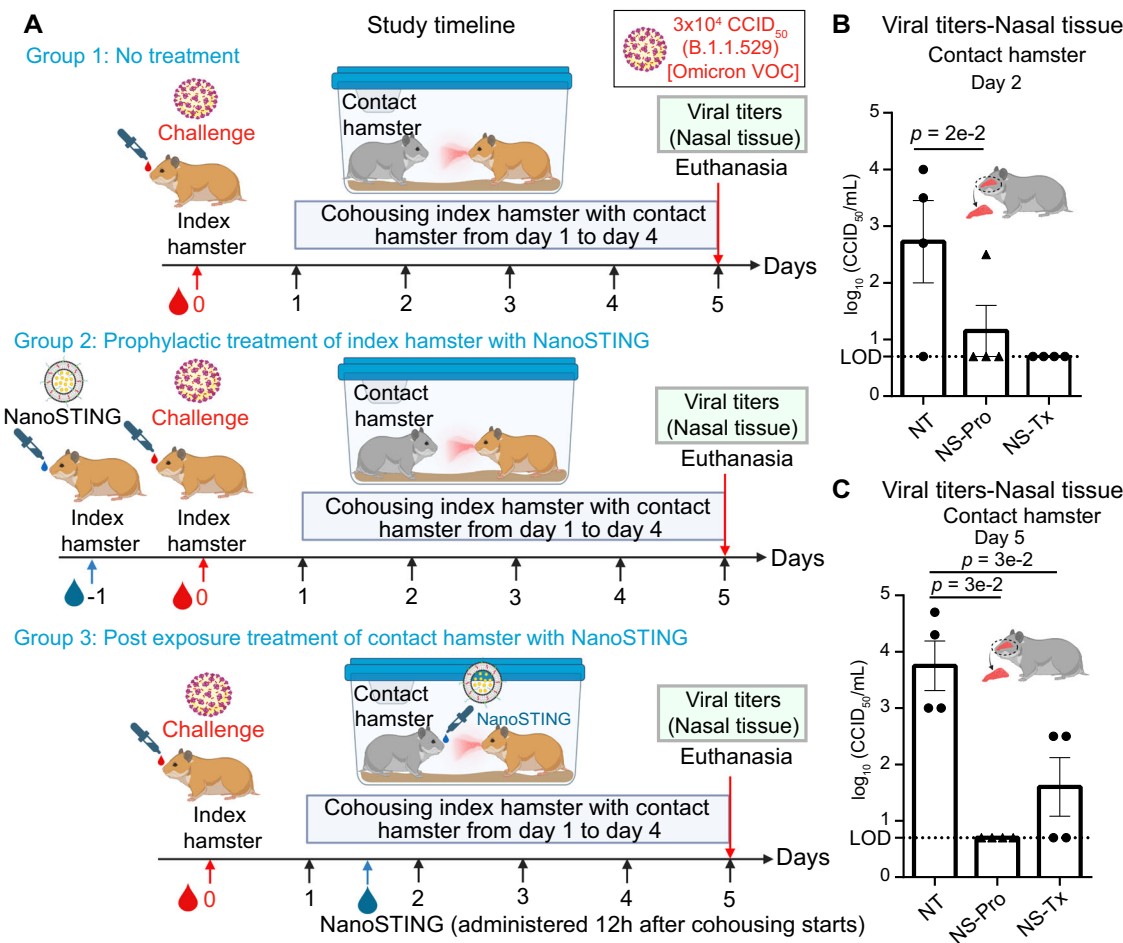

**Fig. 8 | Intranasal administration of NanoSTING limits transmission and viral replication in the nasal passage of contact hamsters exposed to the SARS-CoV-2 Omicron (B.1.1.529) VOC. A** Experimental setup: For group 1, we challenged groups of 8 hamsters each on day 0 with ~3 × 10⁴ of SARS-CoV-2 Omicron VOC (B.1.1.529) and after 24 h cohoused index hamsters in pairs with contact hamsters (n = 8) for 4 days in clean cages. In group 2, we pre-treated the hamsters with 120 μg of NanoSTING 24 h prior to infection. In group 3, we treated the contact hamsters with NanoSTING 12 h after the cohousing period began. We euthanized the contact and index hamsters on day 4 of cohousing. Viral titers in the nasal tissue of the index and contact hamsters were used as primary endpoints. **B, C** Infectious viral particles in the nasal tissue of contact hamsters at day 2 and day 5 after viral administration post-infection were measured by endpoint titration assay. The dotted line indicates limit of detection of the assay (LOD). For (**B, C**), analysis was performed using a two-tailed Mann–Whitney U-test. Individual data points represent independent biological replicates taken from separate animals; vertical bars show mean values with error bars representing SEM. Each dot represents an individual hamster. Mann–Whitney test: ****$p < 0.0001$; ***$p < 0.001$; **$p < 0.01$; *$p < 0.05$; ns not significant. Data presented as combined results from one (**B, C**) independent animal experiment. Gender was tested as a variable with an equal number of male and female hamsters included in the study. See supplementary Fig. 13. Abbreviations- NS-Pro: Prophylactic treatment with NanoSTING; NS-Tx-Post-exposure treatment with NanoSTING. **A** and parts of **B, C**–Created with BioRender.com released under a Creative Commons Attribution-NonCommercial-NoDerivs 4.0 International license (https://creativecommons.org/licenses/by-nc-nd/4.0/deed.en) Number of animals: n = 8/group. Source data are provided as a Source Data file.

31 ± 2.0%. By contrast, animals treated with oseltamivir were moderately protected, showing a mean peak weight loss of 21 ± 2.0% (Supplementary Fig. 14B). We next compared prophylaxis with either oseltamivir (two doses of 30 mg/kg/day) or NanoSTING (single dose at 40 μg) followed by challenge with 2 × 10⁴ CCID₅₀ of H1N1dpm (Fig. 9A). Prophylactic administration of oseltamivir was ineffective, as animals in the placebo (mean peak weight loss of 28 ± 1.0%) and oseltamivir-treated (mean peak weight loss 33 ± 3.0%) groups showed marked weight loss (Fig. 9B). By comparison, a single dose of NanoSTING offered strong longitudinal protection from weight loss (mean peak weight loss 15 ± 3%). These results demonstrate that prophylactic treatment with NanoSTING is superior to oseltamivir treatment.

The evolution of resistance to treatment is predictable and common with influenza. A single amino acid mutation (His275Tyr) with neuraminidase has led to oseltamivir-resistant influenza viruses in humans[49]. Since NanoSTING relies on the host's innate immune response and should be effective against treatment-resistant strains, we next evaluated its potency against oseltamivir-resistant influenza A in mice. We treated groups of ten mice with a single intranasal dose of NanoSTING (40 μg) and 24 h later challenged with 2 × 10⁴ CCID₅₀ of influenza A/Hong Kong/2369/2009 (H1N1)-H275Y [A-H275Y] (Fig. 9C). On day 28, we rechallenged the animals and monitored weight loss until day 41. We used changes in body weight and percent survival as primary endpoints, while serum IgG and IgA were used as secondary measures of adaptive immunity. We treated one group of mice with oseltamivir (30 mg/kg/day), twice daily for five days as a control[48]. NanoSTING-treated animals were well-protected from weight loss (mean peak weight loss of 8 ± 4.0%) in comparison to oseltamivir treatment (mean peak weight loss of 32 ± 3.0%) [Fig. 9D, F]. The weight loss in the NanoSTING-treated animals was transient between days 6–10, and outside of this window, the weight loss in the animals was no different from that of unchallenged animals (Fig. 9D, F). By contrast,

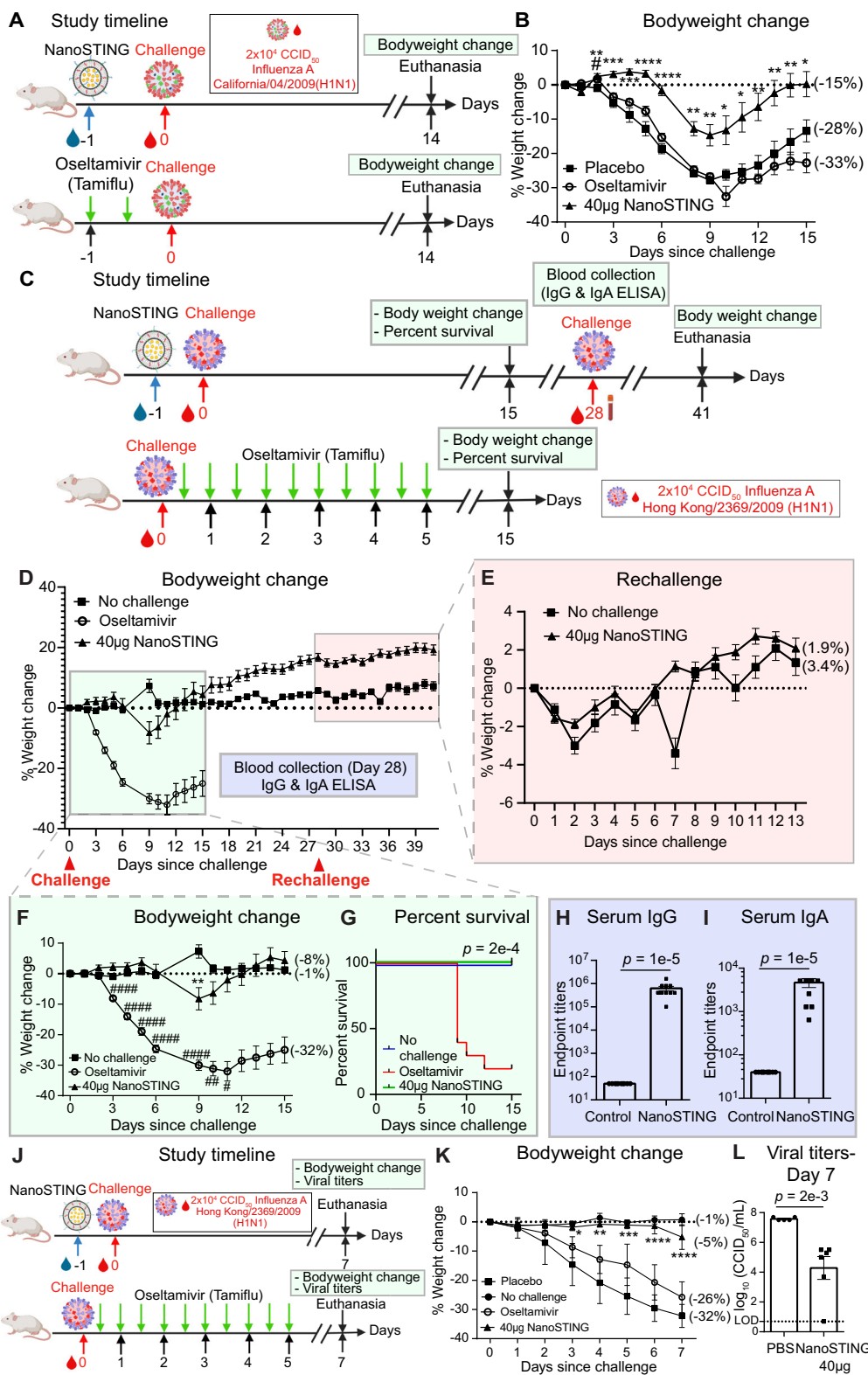

starting at day 4, oseltamivir-treated animals showed significant weight loss until the end of the study (day 15). Consistent with these observations, 100% of NanoSTING-treated animals survived, whereas only 20% of oseltamivir-treated animals survived (Fig. 9G). HA-specific ELISA on day 28 (before rechallenge) confirmed robust IgA and IgG responses (Fig. 9H, I), demonstrating that NanoSTING protected from weight loss without compromising adaptive immunity and

immunological memory. We confirmed that these immune responses are protective; upon rechallenge, the NanoSTING-treated animals were protected from weight loss (mean peak weight loss of 1.9 ± 0.2%) compared to non-challenged animals (mean peak of 3.4 ± 0.4%) [Fig. 9E]. Collectively, these results demonstrate that a single-dose treatment with NanoSTING protects against multiple strains of influenza by establishing immunological memory.

**Fig. 9 | NanoSTING offers protection against Oseltamivir-sensitive and resistant strains of Influenza A. A** Experimental set up: We treated groups of 10 *BALB/c* mice, each with a single dose of NanoSTING (40 µg) or Oseltamivir (30 mg/kg/day administered twice daily) or placebo and 24 h later challenged with $2 \times 10^4$ $CCID_{50}$ of Influenza A/California/04/2009 (H1N1dpm) strain and monitored for 14 days. Body weight change was used as the primary endpoint. Oseltamivir was used as a control. **B** Percent body weight change for the different groups of mice. **C** Experimental set up: We treated groups of 10 *BALB/c* mice with a single intranasal dose of NanoSTING (40 µg) and 24 h later challenged with $2 \times 10^4$ $CCID_{50}$ of influenza A/Hong Kong/2369/2009 (H1N1)-H275Y [A-H275Y] followed by rechallenge on day 28 and tracked the body weight change until day 35. We evaluated the animals for 41 days and used weight loss as the primary endpoint. On day 15, we evaluated the percent survival of different groups of mice. We conducted IgG and IgA ELISA on day 28. We treated one group of mice with a clinically relevant dose of oseltamivir, twice daily for five days. **D** Percent weight change compared to the weight at day 0 at the indicated time intervals. **E** Percent body weight change monitored after rechallenge (day 28–day 41). **F** Percent body weight change monitored during the primary infection (day 0–day 15). **G** Percent survival of the different groups of mice. **H** Humoral immune responses in the serum were evaluated on day 28 using IgG ELISA. **I** Humoral immune responses in the serum were evaluated on day 28 using IgA ELISA. **J** Experimental set up: We treated groups of 10 *BALB/c* mice with a single intranasal dose of NanoSTING (40 µg), and 24 h later challenged with $2 \times 10^4$ $CCID_{50}$ of influenza A/Hong Kong/2369/2009 (H1N1)-H275Y [A-H275Y]. We monitored the animals for 7 days for body weight change and quantified viral titers at the end of the study. We treated one group of mice with oseltamivir, twice daily, for five days. **K** Weight change of the different groups of mice. **L** Viral titers were measured by endpoint titration assay in lungs post 7 days after infection. The dotted line indicates the limit of detection of the assay (LOD). For (**H, I, L**), analysis was performed using a two-tailed Mann–Whitney U-test. Individual data points represent independent biological replicates taken from separate animals; vertical bars show mean values with error bars representing SEM. Each dot represents an individual mouse.

For (**B, F, K**), weight data was compared via a mixed-effects model for repeated measures analysis. Lines depict group mean body weight change from day 0; error bars represent SEM. For (**B, K**), asterisks indicate statistically significant differences between the NanoSTING-treated group and placebo-treated animals, whereas, the pound sign shows statistically significant differences between the Oseltamivir-treated group and placebo-treated animals. For (**F**), asterisks indicate statistically significant differences between the NanoSTING-treated group and non-challenged animals, whereas, pound sign indicate statistically significant differences between the Oseltamivir-treated group and non-challenged animals. For (**G**) we compared survival percentages between NanoSTING-treated and Oseltamivir-treated animals using the Log-Rank Test (Mantel–Cox). ****$p < 0.0001$; ***$p < 0.001$; **$p < 0.01$; *$p < 0.05$; ns not significant. For (**B**) the exact $p$-values comparing the 40 µg NanoSTING group to the Placebo group are as follows: Day 2: $p = 5e{-}3$, Day 3: $p = 5.5e{-}4$, Day 4: $p = 3e{-}4$, Day 5: $p = 2e{-}5$, Day 6: $p = 3e{-}7$, Day: 8: $p = 1e{-}4$, Day 9: $p = 4e{-}3$, Day 10: $p = 2.5e{-}2$, Day 11: $p = 1e{-}2$, Day 12: $p = 6e{-}3$, Day 13: $p = 6e{-}3$, Day 14: $p = 8e{-}3$ and Day 15: $p = 2e{-}2$. Additionally, for the Oseltamivir and Placebo-treated group, the p-values are as follows: Day 2: $p = 1e{-}2$. For (**F**) the exact p-values comparing the 40 µg NanoSTING group to the Placebo group are as follows: Day 9: $p = 5e{-}3$. Additionally, for the Oseltamivir and Placebo-treated group, the p-values are as follows: Day 3: $p = 6e{-}6$, Day 4: $p = 2e{-}8$, Day 5: $p = 10e{-}10$, Day 6: $p = 2e{-}10$, Day 9: $p = 6e{-}8$, Day 10: $p = 1e{-}3$, Day 11: $p = 1e{-}2$. For (**K**) the exact $p$-values comparing the 40 µg NanoSTING group to the Placebo group are as follows: Day 3: $p = 4e{-}2$, Day 4: $p = 5e{-}3$, Day 5: $p = 9e{-}4$, Day 6: $p = 7e{-}6$, Day 7: $p = 1e{-}5$. The data combines results from three independent animal studies: Study 1 (**A, B**), Study 2 (**C–I**), and Study 3 (**J–L**), each involving one independent experiment. Gender was tested as a variable with an equal number of male and female mice included in the study. See also Supplementary Fig. 14. **A, J**–Created with BioRender.com released under a Creative Commons Attribution-NonCommercial-NoDerivs 4.0 International license (https://creativecommons.org/licenses/by-nc-nd/4.0/deed.en) Number of animals used: n = 10/group. Source data are provided as a Source Data file.

To test the impact of NanoSTING treatment on viral titers within the lung, we repeated the challenge experiments with influenza A-H275Y and euthanized the animals on day 7 (Fig. 9J). A single-dose treatment with NanoSTING again protected animals from weight loss (mean peak weight loss of 5 ± 2.0% vs. 32 ± 2.0% placebo) [Fig. 9K]. Infectious viral particles in the lung 7 days after viral exposure were reduced by 500-fold compared to the placebo-treated group, accounting for the ability of NanoSTING to help prevent disease and death (Fig. 9L). In aggregate, these experiments confirmed that NanoSTING works as a broad-spectrum antiviral against influenza by protecting from weight loss, reducing viral titers, and preventing death.

### NanoSTING activates innate immunity in upper airways in Rhesus macaques
To assess the impact of NanoSTING on Rhesus macaques (*M. mulatta*), we administered intranasally four animals with two doses of NanoSTING (0.1 mg/kg-range: 0.06–0.14 mg/kg) on day 0 and day 2. We monitored the animals for four days to track changes in body weight, attitude, appetite, body temperature, and temperature of the nasal cavity (Fig. 10A). Based on our mice studies, we prioritized the measurement of CXCL10 to quantify the activation of innate immunity by NanoSTING. Accordingly, we performed a simple saline wash to collect the nasal fluid for assessments of CXCL10. None of the animals showed clinical signs such as loss of body weight (Fig. 10B) or an increase in body temperature (Fig. 10C) upon administration of NanoSTING. We recorded the temperature for the entire nasal area (Fig. 10D) and right/left nasal areas (Supplementary Fig. 15A, B) before and after NanoSTING administration with a typical facial thermogram. We saw no significant increase in nasal temperatures upon delivery of NanoSTING. ELISA measurements confirmed that we saw a significant increase in the concentration of CXCL10 in the nasal washes at 24 h after administration, and this was reset to baseline at 48 h (Fig. 10E). Repeat-dose administration of NanoSTING increased the concentration of CXLC10,

similar to the effect mediated by the first dose (Fig. 10E). We euthanized one of these treated animals and collected the trachea and lungs. These tissues were processed for routine Hematoxylin and Eosin staining. Histopathological evaluation of the lungs and trachea was unremarkable, providing direct evidence that NanoSTING can safely activate innate immunity (Fig. 10F, G, and Supplementary Fig. 16).

## Discussion
The availability of prophylactic and post-exposure treatments that can prevent disease and reduce transmission of viruses is an urgent and unmet clinical need. Here, we have demonstrated that a single dose of intranasal NanoSTING can work as prophylactic and therapeutic against multiple respiratory viruses (and standard treatment-resistant variants).

The current pandemic has once again highlighted that our therapeutic arsenal against RNA viruses is inadequate. Vaccines are our preferred means of protection against SARS-CoV-2, but they suffer from three drawbacks. First, while the current generation of vaccines was tested at remarkable speed, even this rate of development lags as vaccines need to be custom-manufactured for each emerging virus. Second, the mutational plasticity of RNA viruses like SARS-CoV-2 facilitates their evolution, and newer variants with immune escape potential have emerged. This necessitates ongoing booster shots in adults to achieve at least transitory, complete protection from disease, even as the entire human population is not yet fully vaccinated against SARS-CoV-2[50,51]. As the human experience with influenza has illustrated, requiring additional booster shots reduces human compliance, facilitating the spread of disease. Compounding this problem is that immunosuppressed vaccine recipients fail to be sufficiently protected, and reservoirs are emerging for SARS-CoV-2 outside of humans[52]. Third, despite the efficacy of the current intramuscular vaccines in preventing disease, they do not prevent transmission[53]. The evolution of the SARS-CoV-2 Omicron (B.1.1.529) VOC shows that viruses can quickly adapt to facilitate rapid spread using the nasal

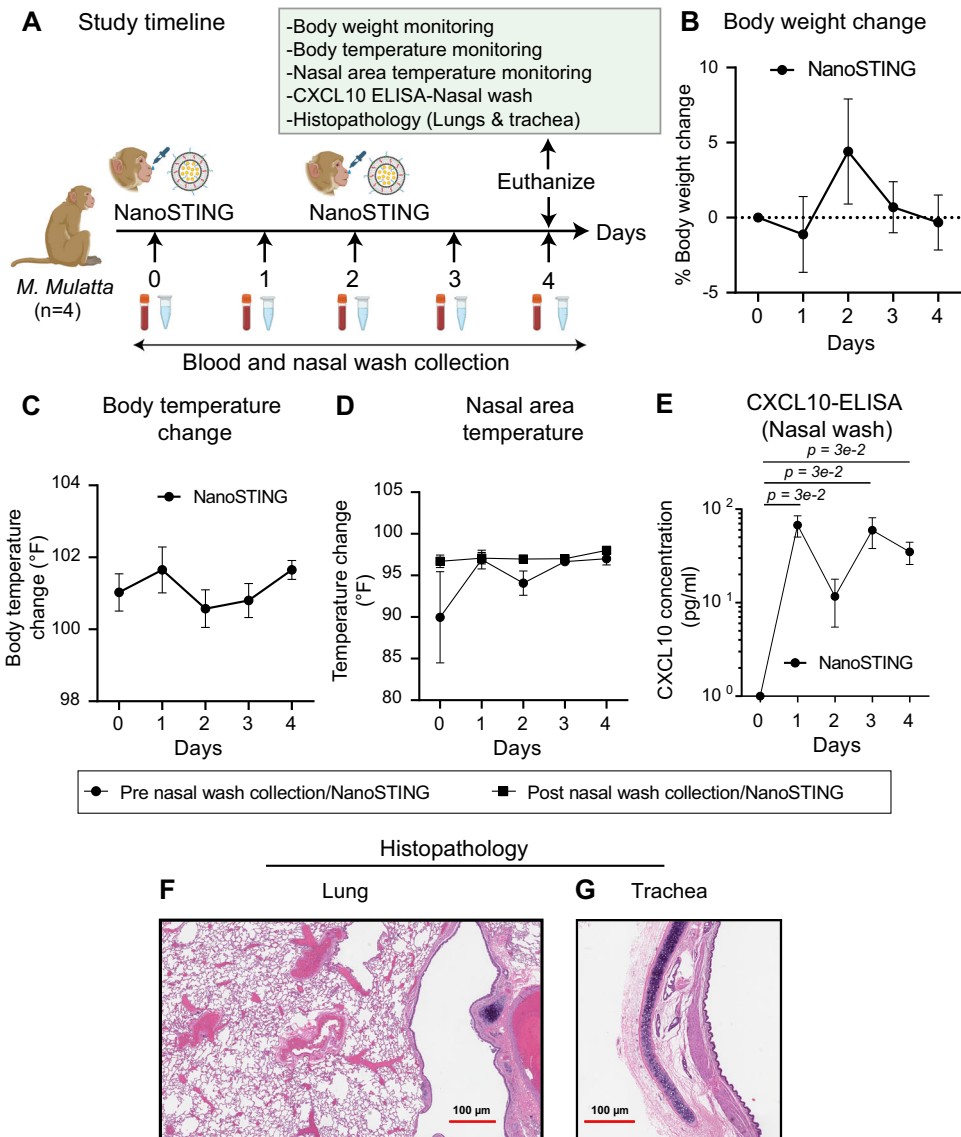

**Fig. 10 | NanoSTING activates innate immunity in upper airways in Rhesus macaques. A** Experimental set up: We administered one group (n = 4/group) of Rhesus macaques (RM's) with two doses of NanoSTING (0.1 mg/kg-range: 0.06–0.14 mg/kg) administered intranasally on day 0 and day 2, and we monitored the animals until day 4 for changes in body weight, body temperature, and nasal area temperature. We euthanized one of the animals on day 4 to assess the histo-pathological changes in the lungs and trachea. **B** Percent body weight change for the RM's at indicated time intervals. **C** Body temperature change for RM's at indi-cated time intervals. **D** Monitoring of nasal area temperature pre and post-nasal wash collection/NanoSTING treatment. **E** Quantification of CXCL10 levels in the nasal wash of animals using quantitative ELISA. **F, G** Representative hematoxylin and eosin (H & E) images of the lungs and trachea of RM's treated with two doses of

NanoSTING (0.1 mg/kg-range: 0.06–0.14 mg/kg); all images were acquired at 2×; scale bar, 100 μm. For (**B, C, D**), the analysis was performed using Kruskal–Wallis test. For (**E**), we performed Mann–Whitney U-test. Individual data points represent independent biological replicates taken from separate animals. Kruskal–Wallis test, Mann–Whitney U-test ****$p < 0.0001$; ***$p < 0.001$; **$p < 0.01$; *$p < 0.05$; ns not sig-nificant. Data presented as combined results from one (**B–G**) independent animal experiment. 3 female and 1 male RM's were taken for the study. See also Supple-mentary Figs. 15 and 16. **A**-Created with BioRender.com released under a Creative Commons Attribution-NonCommercial-NoDerivs 4.0 International license (https://creativecommons.org/licenses/by-nc-nd/4.0/deed.en). Number of animals: n = 4/group Source data are provided as a Source Data file.

cavity as a sanctuary. Thus, while vaccines are necessary, they are not sufficient to fight RNA viruses.

Monoclonal antibodies targeting viruses, like vaccines, offer protection against respiratory disease but suffer from the same dis-advantages as vaccines listed above. Furthermore, their window of use is limited as the emergence of SARS-CoV-2 VOCs such as Omicron can quickly render them ineffective[50]. Additionally, monoclonal antibodies are expensive and administered in a clinical setting, limiting their widespread use. NanoSTING offers an alternative by generating an immune response that appears advantageous for instilling immunity. Focusing on efficacy, intravenous prophylactic administration of

antibody (12 h before challenge) in hamsters led to protection from surrogates of clinical disease (∼2–5% weight loss and ∼300-fold reduction in viral titers in the lung) albeit with no impact on transmission[54,55]. NanoSTING provides a broader window of adminis-tration (24–72 h), with comparable efficacy in reducing clinical disease surrogates while reducing transmission.

Oral antivirals that directly inhibit one or more viral proteins have been developed against SARS-CoV-2 (e.g., paxlovid and molnupiravir) and Influenza (Oseltamivir) are approved for use in humans but are also susceptible to viral evolution and resistance[8]. Furthermore, these therapeutics are designed as an oral post-exposure treatment to

prevent clinical disease and have no impact on viral transmission[56]. In contrast to these pathogen-specific drugs, NanoSTING works broadly against multiple respiratory viruses, including oseltamivir-resistant influenza, highlighting its translational potential. Although we have not undertaken a direct comparative study, based on a review of the literature, the efficacy of NanoSTING compares favorably to the results with paxlovid and molnupiravir in small animal models. The fact that these antivirals are efficacious in humans (30–89% in reducing clinical disease with SARS-CoV-2) suggests that NanoSTING has promising clinical potential[57].

Immunomodulators, including defective viral genome particles, cytokines, and small-molecule agonists, have been tested as antivirals. Defective interfering particles (DIPs) have incomplete genomes and, when administered therapeutically, inhibit replication of the wild-type virus[58]. Although these particles have demonstrated efficacy for SARS-CoV-2 and Influenza in mitigating disease in small animal models, the DIPs must be generated for each virus individually[58,59]. Defective viral genomes (DVGs) based on the poliovirus induce a broad IFN-I response and are protective against multiple viruses[60]. However, DVGs need to replicate in vivo after administration, and this limited replication is essential for their efficacy. However, their broad applicability is limited by concerns about both safety and the presence of pre-existing antibodies in vaccinated people. Lipid nanoparticles complexed with the defective genomes can mitigate these concerns and have shown efficacy against SARS-CoV-2 VOCs in K18-hACE2 mice; the generalizability of this approach in the absence of viral replication to other viruses has not been demonstrated[60].

Direct administration of aerosolized interferons to engage antiviral innate immunity has been tested in animals and humans. In hamsters challenged with SARS-CoV-2, prophylactic or early administration of universal IFN reduces lung damage, provides moderate protection against weight loss (10% vs. 20% for untreated animals), and reduces infectious viral particles (100-fold)[61,62]. NanoSTING appears to offer superior efficacy when compared to these data. In humans, post-exposure treatment with nebulized IFN-α2b was associated with reduced in-hospital mortality compared to no administration of IFN-α2b. By contrast, administration of IFN-α2b more than five days after admission delayed recovery and increased mortality, suggesting that the timing of IFN-α2b administration is critical for efficacy[63]. The limited impact of IFN-α for COVID-19 mirrors its negligible efficacy as a prophylactic against Influenza in humans[64]. In comparison to nebulized interferons, intranasal administration of NanoSTING yields sustained but localized activation of interferons. In combination with the repeat-dose safety data and the in vitro stability data, intranasal NanoSTING thus provides a promising translational path.

Other synthetic small-molecule agonists of pattern recognition receptors (PRRs), including stem-loop RNA 14 (SLR14), a minimal RIG-I (Retinoic acid-inducible gene I) agonist, and STING agonist, diAbzl, have been tested against SARS-CoV-2 in K18-hACE2 mice[22,44,65,66]. A pair of recent studies specifically highlight the efficacy of diABZI-4 in inducing protective innate immune responses against SARS-CoV-2 in murine models, further emphasizing the potential of STING-mediated defenses against viral infections[65,66]. As with all small-molecule drugs, the safety, off-target activity, and pharmacokinetics of synthetic STING agonists must be thoroughly evaluated before translation. NanoSTING is comprised of naturally occurring lipids that have already been tested in humans and cGAMP, the immunotransmitter of danger signals that are conserved across mammals, including humans[67]. As illustrated, NanoSTING leads to safe and sustained delivery and consequently functions as a broad-spectrum antiviral.

Our data illustrate that NanoSTING is a promising immune activator, is safe, stable, and effective against multiple viruses and variants, and can activate innate immunity in non-human primates. It achieves its antiviral effects by rapidly engaging and sustaining activation of the STING pathway. Indeed, an advantage of using the natural

immunotransmitter, cGAMP is that STING activation can lead to both IFN-dependent and independent activities to control viral replication[19,20,67]. NanoSTING exhibits a broad spectrum of activity against existing viruses; activating the innate response protects against current viruses and likely emerging threats. Furthermore, in animal models, NanoSTING minimizes clinical symptoms during primary infection while preserving durable protection from reinfection by eliciting immunological memory. This offers the potential to protect the host from secondary challenges without the need for retreatment. We envision intranasal NanoSTING as a treatment to prevent respiratory viral disease in vulnerable populations or to intervene in respiratory infections before etiology is determined rapidly.

## Methods

### Preparation of NanoSTING

The liposomes contained DPPC, DPPG, Cholesterol (Chol), and DPPE-PEG2000 (Avanti Polar lipids) in a molar ratio of 10:1:1:1. To prepare the liposomes, we mixed the lipids in $CH_3OH$ and $CHCl_3$, and we evaporated them at 45 °C using a vacuum rotary evaporator. The resulting lipid thin film was dried in a hood to remove residual organic solvent. We hydrated the lipid film by adding a pre-warmed cGAMP (MedChemExpress) solution (3 mg/mL in PBS buffer at pH 7.4). We mixed the hydrated lipids at an elevated temperature of 65 °C for an additional 30 min and then subjected them to freeze-thaw cycles. Next, we sonicated the mixture for 60 min using a Branson Sonicator (40 kHz) and used Amicon Ultrafiltration units (MW cut off 10 kDa) to remove the free untrapped cGAMP. Finally, we washed the NanoSTING (liposomally encapsulated STINGa) three times using PBS buffer. We measured the cGAMP concentration in the filtrates against a calibration curve of cGAMP at 260 nm using the Take3 Micro-Volume absorbance analyzer of Cytation 5 (Bio-Tek). We calculated the final concentration of cGAMP in NanoSTING and encapsulation efficiency by subtracting the concentration of free drug in the filtrate.

To check the stability, we stored the NanoSTING at 24 °C and 37 °C for 1, 2, 3, 7, 14, and 30 days. We measured the average hydrodynamic diameter and zeta potential of liposomal particles using DLS and a zeta sizer on Litesizer 500 (Anton Paar).

### Cell lines

THP-1 dual cell line (human, Invivogen: Cat No. thpd-nfis) was cultured in a humidified incubator at 37 °C and 5% $CO_2$ and grown in RPMI/10% FBS (Corning, NY, USA). In addition, we supplemented the THP-1 dual cell line with the respective selection agents (100 µg/mL zeocin + 10 µg/mL blasticidin) and the corresponding selection cytostatics from Invivogen.

### Cell stimulation experiments with luciferase reporter enzyme detection

We performed the cell stimulation experiments using the manufacturer's instructions (Invivogen, CA, USA). First, we seeded the cells in a 96-well plate at $1 \times 10^5$ cells/well in 180 µL growth medium. Next, we made serial dilutions of NanoSTING on a separate plate at concentrations ranging from 2.5 to 10 µg/mL in the growth medium. We then incubated the cells at 37 °C for 24 h. To detect IRF activity, we collected 10 µL of culture supernatant/well at 6 h, 12 h, and 24 h and transferred it to a white (opaque) 96-well plate. Next, we read the plate on Cytation 7 (Cytation 7, Bio-Tek Instruments, Inc.) after adding 50 µL QUANTI-Luc™ (Invivogen) substrate solution per well, followed by immediate luminescence measurement. The data was recorded as relative light units (RLU).

### Viability assessment of cGAMP and NanoSTING

THP-1 dual cells were resuspended in complete RPMI 1640 media supplemented with 10% FBS and 100 nM SYTOX green (Invitrogen, cat.

# S34860). They were incubated on a 96-well plate at a density of 1,00,000 cells per well at 37 °C and 5% $CO_2$. The cells were then stimulated with 2.5–10.0 μg of either cGAMP or NanoSTING. Wells containing unstimulated cells were used as control samples. The cells were imaged using a Cytation 7 inverted microscope using 20× Plan Fluorite WD 6.6 NA 0.45 objective from the FITC and Brightfield channels at 1 h and 12 h post-stimulation.

We detected and counted the number of dead cells and the total number of cells for each time point by segmenting objects from corresponding GFP and bright field images. For GFP images, we applied blob detection with Laplacian of Gaussian as kernel to detect the bright blobs and an overlapping threshold of 0.5. We picked the detections with a 1.2–5 μm radius and a Gaussian sigma value higher than 0.1, each representing one dead cell. On the other hand, for bright field images, we used the CellPose[68] model to segment the cells. Specifically, we used the weights pre-trained by the original paper and followed the default setting of the CellPose detection workflow, except for increasing the flow error threshold to 0.8. As a result, we obtained individual detections and the number of cells for both GFP and bright field images. We calculated the percentage of dead cells as the number of GPF-positive cells normalized to the total number of cells at each time point and plotted the change in the percentage of dead cells after 12 h of stimulation.

## Mice and NanoSTING treatment
All studies using animal experiments were reviewed and approved by the University of Houston (UH) IACUC. We purchased the female 7–9-week-old *BALB/c* mice from Charles River Laboratories (Strain code: 028). The mice were maintained within a Specific Pathogen-Free (SPF) facility housed on ventilated racks within microisolation caging systems. Notably, the mice were not bred within the facility premises and were cohoused during the study. The housing facility for mice was under a 12:12-h light: dark cycle at temperatures 20–22 °C, humidity 40–50%. After sedating them with isoflurane, we intranasally treated the groups of *BALB/c* mice (n = 3–12/group) with varying amounts of NanoSTING (10–40 μg). We euthanized the animals by cervical dislocation after 6 h, 12 h, 24 h, 36 h, and 48 h and harvested blood, nasal turbinates, and lungs. We kept the blood at room temperature (RT) for 10 min to facilitate clotting and centrifuged it for 5 min at 2000 × g. We collected the serum, stored it at −80 °C, and used it for ELISA.

## ELISA
We homogenized nasal turbinates and lung tissue samples in 1:20 (w/v) of tissue protein extraction reagent (Thermo Fisher, # 78510), then centrifuged them for 10 min at 2500 × g to pellet tissue debris. Using quantitative ELISA, we assayed the supernatants for cGAMP, IFN-β, and CXCL10. cGAMP ELISA was performed using a 2'3'-cGAMP ELISA kit (Cayman Chemicals, MI, USA). IFN-β concentrations were tested using a mouse IFN-beta Quantikine ELISA kit (R&D Systems, MN, USA). Mouse IP-10 ELISA kit (CXCL10) was used to perform the CXCL10 ELISAs (Abcam, MA, USA). cGAMP, IFN-β, and CXCL10 concentrations were tested by titering 30 μg total protein from nasal turbinates and lung lysates. All serum samples were tested at 50× dilutions to test cGAMP, IFN-β, and CXCL10 concentrations.

## RNA isolation, cDNA preparation, and qRT-PCR
We excised mouse nasal turbinate tissues and placed approximately 20 mg of tissue in 2 mL tubes containing 500 μL RNeasy lysis buffer (RLT) and a single stainless-steel bead. Next, we homogenized the tissue using a tissue lyser (Qiagen, Hilden, Germany) before total RNA extraction using an RNeasy kit (Qiagen, #74104), following the manufacturer's instructions. Extracted RNA was treated with DNase using a DNA-free DNA removal kit (Invitrogen, #AM1906). Next, 1 μg of total RNA was converted to cDNA using a High-Capacity cDNA reverse transcription kit (Invitrogen, #4368813). We diluted the resultant

cDNA to 1:10 before analyzing quantitative real-time polymerase chain reaction (qRT-PCR). We performed qRT-PCR reaction using SsoFastTM EvaGreen® Supermix with Low ROX (Biorad, # 1725211) on AriaMx Real-time PCR System (Agilent Technologies, Santa Clara, CA). We normalized the results to GAPDH (glyceraldehyde-3-phosphate dehydrogenase). We determined the fold change using the 2-DDCt method, comparing treated mice to naive controls. See Supplementary Table 2 for the primer sequences used in this study.

## Single and repeat-dose toxicology study in rats
We intranasally treated two groups (n = 12) of *Sprague Dawley* rats (10–12 weeks; Charles River; Strain code: 001) with a single dose of either 50 μg or 250 μg of NanoSTING. At 24 h after administration, a necropsy was performed. Similarly, we intranasally treated a group of 12 rats with four doses of NanoSTING (Days 1, 4, 7, 10), and a control group (n = 6) was used as a control group. An equal number of male and female rats were used in this study. A panel of toxicokinetics (TK)/Pharmacodynamics (PD), clinical pathology, and histopathology was performed by Product Safety Labs (New Jersey, USA) [Supplementary Tables 5–12].

## Syrian golden hamsters
All studies using animal experiments were reviewed and approved by UH IACUC. We purchased the 6–10 week-old male and female hamsters *(Mesocricetus auratus)* from Charles River Laboratories (Strain code: 049). The hamsters were not bred on-site. All hamsters were singly housed while at the facility.

## Safety studies of NanoSTING on Syrian golden hamsters
We designed a pilot study to test whether repeated NanoSTING administration causes clinical symptoms (fever or weight loss). We administered a group (n = 4/group) of animals with daily doses of 60 μg of NanoSTING intranasally for four consecutive days. We used naive hamsters as controls (n = 4/group). The animals were monitored daily for body weight change and body temperature. We euthanized using $CO_2$ euthanasia the animals 24 h after administering the last dose and harvested lungs.

## Processing of the hamster's lungs for qRT-PCR and mRNA sequencing
Each lung was cut into 100–300 $mm^2$ pieces using a scalpel to isolate single-cell suspension from the lungs. We transferred the minced tissue to a tube containing 5 mL of digestion Buffer containing collagenase D (2 mg/mL, Roche #11088858001) and DNase (0.125 mg/mL, Sigma #DN25) in 5 mL of RPMI (Corning, NY, USA) for 1 h and 30 min at 37 °C in the water bath with vortexing after every 10 min. We disrupted the remaining intact tissue by passaging (6–8 times) through a 21-gauge needle. After 1 h and 30 min of incubation, we added 500 μL of iced-stopping buffer (1× PBS, 0.1 M EDTA) to each falcon tube to stop the reaction. We then removed tissue fragments and the majority of the dead cells with a 40 μm disposable cell strainer (Falcon, #352340), and we collected the cells after centrifugation. We lysed the red blood cells by resuspending the cell pellet in 3 mL of ACK lysing Buffer (Gibco, #A1049201) and incubated for 3 min at RT, followed by centrifugation. We discarded the supernatants and resuspended the cell pellets in 5 mL of complete RPMI medium (Corning, NY, USA). We enumerated lung cells by trypan blue exclusion.

## qRT-PCR and mRNA sequencing for hamster's lung cells
Total RNA was extracted from whole lung cells using an RNeasy kit (Qiagen, #74104), following the manufacturer's instructions. Extracted RNA was treated with DNase using a DNA-free DNA removal kit (Invitrogen, #AM1906). 1 μg of total RNA was converted to cDNA using a High-capacity cDNA reverse transcription kit (Invitrogen, #4368813). We diluted the resultant cDNA to 1:10 for qRT-PCR. We performed

qRT-PCR reaction using SsoFastTM EvaGreen® Supermix with Low ROX (Biorad, #1725211) on AriaMx Real-time PCR System (Agilent Technologies, Santa Clara, CA). We normalized the results to *Actb* (β-actin gene). We determined the fold change using the 2-DDCt method, comparing treated mice to naive controls. The primer sequences are provided in Supplementary Table 3. The preparation of the RNA library and mRNA sequencing was conducted by Novogene Co., LTD (Beijing, China). We paired and trimmed the fastq files using Trimmomatic (v 0.39) and aligned them to the Syrian golden hamster genome (MesAur 1.0, ensembl) using STAR (v 2.7.9a). We determined the differential gene expression using DESeq2 (v 1.28.1) package[69]. To perform gene set enrichment analysis, we used a pre-ranked gene list of differentially expressed genes in GSEA software (UC San Diego and Broad Institute). To generate the gene set for IFN-independent activities of STING, we collected genes with a 2-fold change increase in BMDM-STING S365A-DMXAA vs BMDM-STING S365A-DMSO samples from the GSE149744 dataset as described previously[19].

## Preparation of DiD-SRB-loaded liposomes

We used liposomes composed of a molar ratio of 10:1:1:1 of DPPC, DPPG, Cholesterol (Chol), and DPPE-PEG2000. We added DiD to the lipid mixture with a 0.5 µmol/mL concentration. To prepare the liposomes, we mixed the lipids (16.9 mg of DPPC, 1.8 mg of DPPG, 0.9 mg of cholesterol, and 6.4 mg of DPPE-PEG2000) in 0.85 mL of chloroform and 0.341 mL of methanol in a round bottom flask. We vortexed this mixture to dissolve the lipids in the solvent solution. We made a stock of DiD solution. We added 1 gram of DiD to 50 mL of methanol and vortexed thoroughly. Next, we added 0.025 mL of this stock to the lipid/solvent mixture, then evaporated the solvents in the lipid mixture using a rotary evaporator (for 1 h) to form a lipid film. Next, we hydrated the lipid film by adding a pre-warmed 1 ml of SRB solution (50 mg/mL in PBS). Immediately after adding the SRB solution, the hydrated film should be vigorously vortexed. We mixed the hydrated lipids in a water bath at an elevated temperature of 65 °C (or a temperature above the transition temperature of the lipids) for an additional 30 min with vigorous vortexing every 5 min. The mixture was subjected to 10 freeze-thaw cycles by cooling it to −80 °C and warming it to RT (~25 °C). Next, we extruded the mixture with an Avanti extruder kit at 65 °C using a 0.2 µm pore filter. The mixture should be passed through the filter 10 times or more. We performed NTA at this step on the liposomes. The mode of measurement 5 recording sets on the NanoSight NS300 Malvern Panalytical instrument at 10,000× dilution in milli-Q water. We removed the free untrapped SRB through dialysis (100 kDa membrane) for 24–48 h at 4 °C with continuous stirring and exchanged the PBS dialysate two times with fresh PBS. We characterized the DiD-SRB liposomes with a THP-1 assay and endotoxin assay.

## Intranasal dosing of DiD-SRB liposomes to mice

We dosed a group of five *BALB/c* mice intranasally with DiD-SRB liposomes and euthanized the animals by cervical dislocation post 12 h. We harvested lungs and nasal tissues from indicated mice and processed them into single-cell suspensions for analysis by flow cytometry.

## Tissue processing post DiD-SRB liposomes administration to mice

To isolate lung cells, we perfused the lung vasculature with 5 ml of 1 mM EDTA in PBS without $Ca^{2+}$, $Mg^{2+}$ and injected it into the right cardiac ventricle. Nasal tissue and lung were cut into 100–300 mm² pieces using a scalpel. We transferred the minced tissue to a tube containing 5 ml of digestion buffer containing collagenase D (2 mg/ml, Roche #11088858001) and DNase (0.125 mg/ml, Sigma #DN25) in 5 ml of RPMI-1650 for 1 h and 30 min at 37 °C in the water bath by vortexing after every 10 min. We disrupted the remaining intact tissue by passage (6–8 times) through a 21-gauge needle. Next, we added 500 µL of ice

cold-stopping buffer (1× PBS, 0.1 M EDTA) to stop the reaction. We then removed tissue fragments and dead cells with a 40 µm disposable cell strainer (Falcon) and collected the cells after centrifugation at $400 \times g$. We then lysed the red blood cells (RBCs) by resuspending the cell pellet in 3 ml of ACK Lysing Buffer (Invitrogen) and incubated for 3 min at RT, followed by centrifugation for 10 min at $400 \times g$. Then, we discarded the supernatants and resuspended the cell pellets in 5 ml of complete RPMI medium (Corning, NY, USA). Using the trypan blue exclusion method, we counted the lung and nasal tissue cells.

## Cell surface staining for flow cytometry

We collected the cells and stained them with Live/Dead Aqua (Thermo Fisher #L34965) in PBS, followed by Fc-receptor blockade with anti-CD16/CD32 (Thermo Fisher #14-0161-85), and then stained for 30 min on ice with the following flourescent labeled antibodies/conjugates in flow cytometry staining buffer (FACS): anti-CD45, anti-EPCAM, anti-CD31, anti-CD11b, anti-CD11c, anti-CD24 and GS-IB$_4$ conjugate. We washed the cells twice with the FACS buffer and analyzed them on LSR-Fortessa flow cytometer (BD Bioscience) using FlowJo™ software version 10.8 (Tree Star Inc, Ashland, OR, USA). Cell populations and subsets in the mouse respiratory system were gated and analyzed as described[27]. Information on various antibodies and conjugates and the dilution used is provided in Supplementary Table 4. See Supplementary Fig. 5 for the gating strategy.

## Viruses

Isolates of SARS-CoV-2 were obtained from BEI Resources (Manassas, VA) and amplified in Vero E6 cells to create working stocks of the virus. Influenza A/California/04/2009 was kindly provided by Elena Govorkova (St. Jude Children's Research Hospital, Memphis, TN) and was adapted to mice by Natalia Ilyushina and colleagues at the same institution. Influenza A/Hong Kong/2369/2009 (H1N1pdm) was provided by Kwok-Yung Yuen from The University of Hong Kong, Hong Kong Special Administrative Region, People's Republic of China. The virus was adapted to mice by four serial passages in the lungs of mice, and plaque was purified at USU.

## Biosafety

Studies with influenza virus were completed within the ABSL-2 space of the Laboratory Animal Research Center (LARC) at USU. Studies involving SARS-CoV-2 were completed within the ABSL-3 space of the LARC at USU.

## Transmission studies

For group 1, we challenged groups of five hamsters each on day 0 with $\sim 3 \times 10^4$ of SARS-CoV-2 Omicron VOC (BA.5) and after 24 h cohoused index hamsters in pairs with contact hamsters (n = 5) for four days in clean cages. In group 2, we pre-treated the hamsters with 120 µg of NanoSTING 24 h prior to infection. In group 3, we treated the contact hamsters with NanoSTING 12 h after the cohousing period began. We repeated this study with another strain of Omicron VOC (B.1.1.529). Viral titers in the nasal tissue of the index and contact hamsters were used as primary endpoints. Infectious viral particles in the nasal tissue of contact hamsters on day 2 and day 5 after viral administration post-infection were measured by endpoint titration assay.

## Viral challenge studies in animals

Animals. For SARS-COV-2 animal studies completed at USU, 6–10 week-old male and female golden Syrian hamsters (Strain code: 049) were purchased from Charles River Laboratories and housed in the ABSL-3 animal space within the LARC. For influenza virus animal studies, 8-week-old *BALB/c* (Strain code: 028) mice were purchased from Charles River Laboratories.

Infection of animals. Hamsters were anesthetized with isoflurane and infected by intranasal instillation of $1 \times 10^{4.5}$ CCID$_{50}$ of SARS-CoV-2

in a 100 μl volume. Mice were also anesthetized with isoflurane and infected with a $1 \times 10^{4.3}$ CCID$_{50}$ dose of influenza virus in a 90 μl volume.

Titration of tissue samples. Lung and nasal tissue samples from hamsters and lung tissue samples from mice were homogenized using a bead-mill homogenizer using minimum essential media. Homogenized tissue samples were serially diluted in a test medium and the virus was quantified using an endpoint dilution assay on Vero E6 cells [African green monkey kidney cells-Vero E6 (ATCC®, cat# CRL-1586′M)] for SARS-CoV-2 and on MDCK [Madin-Darby canine kidney- MDCK cells (ATCC®, CCL-34)] cells for influenza virus. A 50% cell culture infectious dose was determined using the Reed-Muench equation[70].

## Safety study in Rhesus macaques (RM's)
Experiments with rhesus macaques (*M. mulatta*) were reviewed and approved by UH IACUC. Four healthy rhesus macaques (RM's) of Indian origin, between 4 and 11 years of age and 4–12 kg in weight were used. The RM's were acquired from Washington University School of Medicine, Division of Comparative Medicine C/O Dr. Chad B Faulkner; 660S. Euclid Ave., Box 8061; St. Louis, MO 63110 and Keeling Center for Comparative Medicine and Research, MD Anderson Cancer Center, Bastrop, TX. We used four RM's for the study. Three of them were males, and one was female. All the animals were single-housed. To assess the impact of NanoSTING on RM's, we administered animals (n=4) with two doses of NanoSTING (700 μg) administered intranasally on day 0 and day 2. The animals were monitored until day 4 for changes in body weight, attitude, appetite, body temperature (via rectal thermometer), and nasal tract temperature using a Veterinary IR Pad 640, a digital thermal infrared camera (Digatherm Veterinary Thermal Imaging, Beaumont, TX). We collected the nasal wash each day for quantification of CXCL10 using quantitative ELISA. One animal was euthanized via intravascular (IV) injection of Euthasol euthanasia solution (Midwest Veterinary Supply). One of the RM's was sedated using 55 mg Ketamine and 0.075 mg Dexmedetomidine (Midwest Veterinary Supply) IM (intramuscularly) before euthanasia on day 4 to assess any histopathological change in the lungs and trachea.

## Histopathology
Lungs of the Syrian golden hamsters and lungs, trachea of Rhesus macaques and small intestines, stomach, lungs, and nasal cavity of rats were fixed in 10% neutral buffered formalin processed, paraffin-embedded, and 4-μm sections were stained with hematoxylin and eosin. We used an integrated scoring rubric for evaluating the pathology score[71]. The published scoring method was modified from a 0–3 to a 0–4 score with 1 = 1–25%; 2 = 26–50%; 3 = 51–75; and 4 = 76–100%. The original histologic criteria comprised of three compartments: airways, blood vessels, and interstitium. The sum of all three scores was reported as the cumulative lung injury score for an animal ranging from 0 to 12. This scoring also takes into account the degeneration/necrosis of the bronchial epithelium/alveolar epithelium. A board-certified pathologist (M.S.) evaluated the sections.

## Quantitative modeling
To quantify the kinetics of SARS-CoV-2 infection in the upper respiratory tract (URT) in the presence of NanoSTING, we modified the innate immune model described by Ke et al.[37]. We added an additional coefficient to the term responsible for refractory responses in the set of governing ordinary differential equations (ODEs), as shown in supplementary information file (Supplementary Figs. 11 and 12). The mean population parameter values and initial values were taken from Ke et al.[37]. We solved the system of ODEs for different efficacies, treatment initiation time, and duration of response of NanoSTING using the ODE45 function in MATLAB 2018b. A sample MATLAB code for solving the system of equations has been provided in Supplementary Note 1.

## Statistics and reproducibility
Statistical significance was assigned when *P* values were <0.05 using GraphPad Prism (v6.07). Tests, number of animals (n), mean values, statistical comparison groups, and the statistical test used are indicated in the figure legends. No statistical methods were used to pre-determine sample sizes for the in vitro and animal studies. The sample size was determined based on similar studies in this field. Animal studies were randomized. When applicable, technical repeats are specified for each experiment in the figure legends wherever applicable. Reproducibility between animals in treatment and naïve controls/placebo-treated groups is shown in the results and figure legends. The researchers were not blinded to allocation during experiments and outcome assessment. Data collection and analysis were not performed blind to the conditions of the experiments. The pathologists performing the histopathological analysis were blinded to treatment. The formulation was manufactured at UH and shipped to USU. All animal experiments at USU were performed independently. Further information on research design is available in the Nature Research Reporting Summary linked to this article.

## Modeling SARS-CoV-2 infection
To quantify the kinetics of SARS-CoV-2 infection in the upper respiratory tract (URT) in the presence of NanoSTING, we used the innate immune model described by Ke et al.[37]. Assuming that NanoSTING efficacy is primarily due to the cell's increased capacity to become refractory to infection, we modified the governing equations, as shown in the Supplementary Table 13 and Fig. 5 of the manuscript.

To get a physical interpretation of the variable NanoSTING, we non dimensionalized the target cell equation in the following way:

$$\frac{\mathrm{d}T}{\mathrm{d}t} = -\beta V T - \varphi I_{max}\left(\frac{\varphi I}{\varphi I_{max}} + \frac{NanoSTING}{\varphi I_{max}}\right)T + \rho R \qquad (1)$$

$$RIR = \frac{NanoSTING}{\varphi I_{max}} \qquad (2)$$

Where RIR is the relative interferon ratio, which is the relative contribution of NanoSTING to antiviral Interferon (refractory) responses compared to peak antiviral Interferon responses during SARS-CoV-2 without NanoSTING.

We solved these ordinary differential equations with mean population parameter values and initial values taken from Ke et. al.[37] and as shown in Supplementary Tables 14 and 15. First, we performed a sensitivity analysis to show that the peak natural SARS-CoV-2 response was independent of initial viral titer (Supplementary Fig. 10A). We also performed a sensitivity analysis to show that NanoSTING was effective at higher viral titers as well (Supplementary Fig. 10B). We calculated the viral titer area under the curve (AUC) during infection for varying RIRs and the treatment initiation time post viral exposure. Because the effect of NanoSTING lasts only for 24–48 h, the NanoSTING coefficient was non-zero only up to 24–48 h post-treatment initiation.

## Reporting summary
Further information on research design is available in the Nature Portfolio Reporting Summary linked to this article.

# Data availability
All data are included in the Supplementary Information or available from the authors, as are unique reagents used in this Article. The raw numbers for charts and graphs are available in the Source Data file whenever possible. Sequencing data reported in this paper has been deposited to GEO (GSE201423) and is publicly available. All material and experimental data requests should be directed to the corresponding author, Navin Varadarajan. Source data are provided with this paper.

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

## Acknowledgements

This publication was supported by the NIH (R01GM143243), Owens Foundation, and AuraVax Therapeutics. We thank Prashant Menon and Kwan Ling Wu for helping with viability assessment studies.

## Author contributions

N.V. conceived the study. N.V., A.L., L.J.N.C., M.S., and B.H. designed the study. A.L., A.S., S.R.S., M.K., M.M.P., R.K., S.B., B.H., H.M., C.M.S., H.M., V.E.D. and X.L. performed experiments. A.L., A.S., S.R.S., M.K., K.R., B.H., X.L., D.T., and N.V. analyzed the data. M.K. performed modeling, and MF performed bioinformatic analyses. N.V. and A.L. drafted the manuscript and all authors contributed to the review and editing of the manuscript.

## Competing interests

UH has filed provisional patents based on the findings of this study. N.V. and L.J.N.C. are co-founders of AuraVax Therapeutics and CellChorus. The remaining authors declare no competing interests.

## Ethics approval

The mouse, hamster, and NHP studies were performed under the study protocol (PROTO2020000019, PROTO202100006, PROTO202100049, PROTO202200025), as approved by the Institutional Animal Care and Use Committee in the University of Houston. The animal experiments at USU were conducted in accordance with an approved protocol by the Institutional Animal Care and Use Committee of Utah State University. The work was performed in the AAALAC-accredited LARC of the university in accordance with the National Institutes of Health Guide for the Care and Use of Laboratory Animals (8th edition; 2011).
