## [Peer Review File · Nature Communications]

An intranasal nanoparticle STING agonist protects against respiratory viruses in animal modelsEditorial Note: This manuscript has been previously reviewed at another journal that is not operating a transparent peer review scheme. This document only contains reviewer comments and rebuttal letters for versions considered at *Nature Communications*.

REVIEWERS' COMMENTS

Reviewer #5 (Remarks to the Author):

The authors responded to the critique of referee #2, second point, with additional data. In the new Fig. 2 they use dye-labelled liposomes to identify Nano-STING targets in the nasal epithelium. The data further improve the manuscript. However, the authors identify macrophages as CD11c+/CD11b-. While this is the case for lung macrophages, shouldn't those of the nasal epithelium be CD11b+/CD11c-?

Point 3 is addressed with an experiment shown for referees, but not intended for publication in this manuscript.

To my opinion the manuscript has been further improved and describes a sufficient amount of novel and important data.

Reviewer #6 (Remarks to the Author):

I have been asked to assess the response of the author's to reviewer #1's comments/concerns. I will attempt to restrict my review to this task (although I have of course read the entire submitted manuscript).

The previous reviewer's primary concern appears to have been re the novelty of the approach versus previous publications of a similar nature. I must admit that I share this concern and while the authors have endeavoured to refer to and in places comment on these prior publications now this by itself does not change the degree of novelty.

Whether there is a clinical application for such a PreP candidate I think can be left up to clinicians and industry and need not be debated at this early stage. The authors have attempted to vary the timing (including pre- and post-exposure) to support the notion that there is a reasonable "window of opportunity" for administration, albeit in healthy, SPF rodents.

I think that the technical concerns/criticisms of reviewer #1 have been adequately satisfied by the authors.

I do think that it needs be rebred that, while tested extensively in rodents here, it may be early to make strong statements re the promise of the approach. Statements like "Our data illustrate that NanoSTING has emerged as a first-in-class immunoantiviral as it is safe, stable, and effective against multiple viruses and variants; and can activate innate immunity in non-human primates." seem overly enthusiastic.

REVIEWERS' COMMENTS

Reviewer #5 (Remarks to the Author):

The authors responded to the critique of referee #2, second point, with additional data. In the new Fig. 2 they use dye-labelled liposomes to identify Nano-STING targets in the nasal epithelium. The data further improve the manuscript. However, the authors identify macrophages as CD11c⁺/CD11b⁻. While this is the case for lung macrophages, shouldn't those of the nasal epithelium be CD11b⁺/CD11c⁻?

Point 3 is addressed with an experiment shown for referees, but not intended for publication in this manuscript.

To my opinion the manuscript has been further improved and describes a sufficient amount of novel and important data.

Response: We agree with your comment and have modified the Fig.2 and showed the population of DID⁺SRB⁺CD45⁺EPCAM⁻CD11b⁺CD11c⁻ and DID⁺SRB⁺CD45⁺EPCAM⁻CD11b⁻CD11c⁺ cells in lungs and nasal compartment.

Reviewer #6 (Remarks to the Author):

I have been asked to assess the response of the author's to reviewer #1's comments/concerns. I will attempt to restrict my review to this task (although I have of course read the entire submitted manuscript).

The previous reviewer's primary concern appears to have been re the novelty of the approach versus previous publications of a similar nature. I must admit that I share this concern and while the authors have endeavoured to refer to and in places comment on these prior publications now this by itself does not change the degree of novelty.

Whether there is a clinical application for such a PreP candidate I think can be left up to clinicians and industry and need not be debated at this early stage. The authors have attempted to vary the timing (including pre- and post-exposure) to support the notion that there is a reasonable "window of opportunity" for administration, albeit in healthy, SPF rodents.

I think that the technical concerns/criticisms of reviewer #1 have been adequately satisfied by the authors.

I do think that it needs be rebred that, while tested extensively in rodents here, it may be early to make strong statements re the promise of the approach. Statements like "Our data

illustrate that NanoSTING has emerged as a first-in-class immunoantiviral as it is safe, stable, and effective against multiple viruses and variants; and can activate innate immunity in non-human primates." seem overly enthusiastic.

Response: Thanks for your feedback. We have revised our manuscript to better reflect a cautious interpretation of our results. We appreciate your insights and believe these changes strengthen the manuscript.